# Development Status of Solar-Driven Interfacial Steam Generation Support Layer Based on Polymers and Biomaterials: A Review

**DOI:** 10.3390/polym16172427

**Published:** 2024-08-27

**Authors:** Haipeng Yan, Pan Wang, Lingsha Li, Zixin Zhao, Yang Xiang, Haoqian Guo, Boli Yang, Xulin Yang, Kui Li, Ying Li, Xiaohong He, Yong You

**Affiliations:** 1School of Mechanical Engineering, Chengdu University, Chengdu 610106, China; yhp200519@foxmail.com (H.Y.); lilingsha@stu.cdu.edu.cn (L.L.); zhao19913609107@163.com (Z.Z.); xiangyang@stu.cdu.cn (Y.X.); jingyu264452@foxmail.com (H.G.); itjbsyday16245791@163.com (B.Y.); yangxulin@cdu.edu.cn (X.Y.); likui@cdu.edu.cn (K.L.); liying@cdu.edu.cn (Y.L.); 2School of Automation, Chengdu University of Information Technology, Chengdu 610225, China; hexiaohong@cuit.edu.cn; 3Key Laboratory of General Chemistry of the National Ethnic Affairs Commission, School of Chemistry and Environment, Southwest Minzu University, Chengdu 610041, China; youyong@swun.edu.cn

**Keywords:** water treatment, solar evaporation, support layer, polymer, biomaterial

## Abstract

With the increasing shortage of water resources and the aggravation of water pollution, solar-driven interfacial steam generation (SISG) technology has garnered considerable attention because of its low energy consumption, simple operation, and environmental friendliness. The popular multi-layer SISG evaporator is composed of two basic structures: a photothermal layer and a support layer. Herein, the support layer underlies the photothermal layer and carries out thermal management, supports the photothermal layer, and transports water to the evaporation interface to improve the stability of the evaporator. While most research focuses on the photothermal layer, the support layer is typically viewed as a supporting object for the photothermal layer. This review focuses on the support layer, which is relatively neglected in evaporator development. It summarizes existing progress in the field of multi-layer interface evaporators, based on various polymers and biomaterials, along with their advantages and disadvantages. Specifically, mainly polymer-based support layers are reviewed, including polymer foams, gels, and their corresponding functional materials, while biomaterial support layers, including natural plants, carbonized biomaterials, and other innovation biomaterials are not. Additionally, the corresponding structure design strategies for the support layer were also involved. It was found that the selection and optimal design of the substrate also played an important role in the efficient operation of the whole steam generation system. Their evolution and refinement are vital for advancing the sustainability and effectiveness of interfacial evaporation technology. The corresponding potential future research direction and application prospects of support layer materials are carefully presented to enable effective responses to global water challenges.

## 1. Introduction

Solar-driven interfacial steam generation (SISG) has emerged as a highly promising strategy for seawater desalination [1]. Different from traditional approaches like reverse osmosis [2,3,4], distillation [5,6,7], and nanofiltration [8,9], photothermal evaporation employs solar energy by converting sunlight irradiation into heat energy, which drives the vapor; the condensed water is collected after evaporation to complete the desalination of sea water. Solar energy is renewable and environmentally friendly, and there is no requirement for large energy consumption, expensive equipment, or high maintenance costs. Generally, as shown in Figure 1, there are three strategies for unitizing solar energy for evaporation [1]: (1) evaporation based on internal heating, in which uniformly dispersed solar absorbers convert incident solar photons into thermal energy to heat the liquid; (2) evaporation based on bottom heating, in which solar energy is absorbed by a solar absorber and converted into thermal energy to heat a large amount of liquid at the bottom of the evaporator; (3) evaporation based on interface heating, where solar thermal conversion and heating are limited to the gas–liquid interface. In recent decades, solar-driven interface evaporation technology has been widely studied; interface water evaporation technology uses solar energy to concentrate heat at the water–air interface for local water evaporation, thus minimizing heat loss in the desalination process [10]. The advanced fabrication of photothermal evaporators, including the employment of reasonable materials and optimized structure design, is the key to realizing efficient interfacial water evaporation.

Common SISG evaporators can be classified into single-layer and multi-layer evaporators. The single-layer evaporator structure typically consists of a lightweight floating layer with photothermal conversion materials directly placed on the water’s surface. Since photothermal conversion materials usually need to be loaded onto a supporting substrate in order to float on the water’s surface, the substrate is usually a porous membrane material, typically a polymer fiber membrane. However, this simplicity often leads to poor durability, heat loss, and other problems (Figure 1) [11,12,13,14].

The multi-layer evaporator typically combines at least a photothermal conversion layer and a support layer. The support layer is usually located at the bottom, providing stable structural support and preventing direct contact between bulk water and the thermal layer, reducing heat loss. It is typically constructed with lightweight materials that can float on the water [15]. The working principle of the multi-layer evaporator is that the top photothermal layer absorbs the incident sunlight and converts it into heat energy. Through capillary action, water gradually infiltrates the photothermal layer from bottom to the top through the water vapor transport channel arranged in the support layer under the photothermal layer, and accumulates into a thin surface water layer at the top of the photothermal layer. The photothermal layer heats the water in the surface water layer, causing it to evaporate into steam. The water in the surface water layer gradually evaporates, and the impurities in the water are retained in the photothermal absorption layer because they cannot evaporate, thus achieving desalination. Multi-layer interfacial evaporators for efficient interfacial evaporation technology are attracting increased attention from researchers for the scientifically based optimization of their parameters such as nature, morphology, thickness, interface, and structure. For example, after years of research, carbon-based materials [16], metal materials [17,18], and semiconductor materials have emerged as good choices for photothermal layer materials. They usually exhibit excellent light absorption and efficient evaporation characteristics. Compared with the core photothermal layer, the support layer has received less attention in many studies. However, with the deepening of research on the support layer, improving the support layer has gradually become an indispensable way to improve the overall performance of an SISG evaporator.

From the increasing number of research reports on SISG evaporators in recent years, it has been found that reasonable material selection and structural design of the support layer can significantly improve evaporation efficiency and structural stability. However, current research often relegates the support layer to a supporting role in discussions, and there remains a lack of comprehensive summarization in this field. This review mainly explores the support layer materials, including various porous polymer materials, biomaterials, and their corresponding composites. In addition, the common optimization of the structural design of the support layer, specifically the new 3D structure (vertebrae and arch), is also presented here. This work aims to describe the development status of the support layer in multi-layer evaporators, and finally propose future research directions for advancing support layer construction in the field of SISG technology for water treatment.

## 2. Support Layer Materials

In order to manage the heat and material loss from direct water surface contact in photothermal evaporators, the typical design involves separating the photothermal layer and support layer. This structure provides thermal insulation protection and structural support. Thus, the thermal conductivity and water absorption characteristics should be carefully considered when choosing the support layer material. At present, most common support layer materials are based on polymers. Among the family of polymer-based support layers, commercial polymer foams are characterized by their excellent thermal insulation and buoyancy [19]. These foams can also be enhanced through modifications to form composite materials, which often exhibit superior properties [20]. Aerogels and hydrogels are also promising support layer materials, and they stand out because of their high polymer crosslinking, facilitating efficient water transport and thermal conductivity [21,22]. At the same time, a support layer fabricated using biomaterials possesses the advantages of a wide range of sources, low cost, and eco-friendliness [23]. Biomaterials offer natural structural advantages, such as porosity and micro-channels, which ensure effective water transport. Some biomaterials even have natural beam tube structures, enhancing their utility as support layer materials. Moreover, reasonable treatments of biomaterials, including drying, carbonization, and modification with functional components, enable them to achieve high evaporation efficiency and suitability for diverse practical applications

### 2.1. Polymer Foam

#### 2.1.1. Commercial Polymer Foam

Polymer foam is a common candidate for support layer fabrication, offering a particularly wide array of choices, such as polyurethane (PU) foams [24,25], polystyrene (PS) foams [3,26], and polydimethylsiloxane (PDMS) foam. These foams are characterized by their light weight, ability to float, and low thermal conductivity, as well as their cost-effectiveness, making them applicable as support layer materials. Their light weight structure allows them to float freely on the water’s surface, which is a critical factor in evaporator design. In addition, foams ensure the stability and reliability of the evaporator. At the same time, foams’ low thermal conductivity effectively insulates against heat loss, thereby preventing the heat generated from sunlight absorption by the photothermal layer from spreading the foam layer. This insulation is responsible for effectively maintaining high evaporation efficiency in the evaporator.

The porous structure of commercial plastic foams is another important feature that confers several advantages during application in SISG technology. The porosity of the foam allows water to easily enter the evaporator and reach the evaporation interface, thereby improving the evaporation efficiency. At the same time, the porous structure provides a larger surface area, allowing more water to absorb and diffuse into capillary water flow, further improving evaporation efficiency. Shi et al. [27] chose polystyrene (PS) foam as the support layer material and covered it with porous graphene oxide (rGO) film in their evaporator design. The top layer of porous rGO acted as a light absorber, efficiently capturing sunlight and converting it into thermal energy. Meanwhile, the bottom PS foam layer effectively breaks down water transport channels and acts as an excellent thermal barrier, minimizing heat transfer to the bulk water. The designed SISG evaporator showed efficient water transfer and a good evaporation efficiency: 1.31 kg m^−2^ h^−1^.

The durability of the polymer foam deteriorates with an increase in the serving time, and the long-term accumulation of heat causes overflow at some point. To address this weakness, Nguyen et al. [28] designed an efficient SISG evaporator device with a polyethylene foam support layer. The durability of the designed evaporator was improved by wrapping polyethylene foam with cotton gauze, thereby minimizing the area of contact with bulk water (Figure 2a). The cotton gauze’s ability to quickly absorb water also ensures the water supply during the evaporation process. Additionally, Kiriarachchi et al. [29] employed a cotton rod structure to raise the foam layer to the surface of the water, preventing direct contact between the water and polymer foam, and facilitating the transfer of water to the top layer for evaporation (Figure 2b).

Due to the increases in evaporation rate achieved in many recent studies, this rate has long surpassed the rate of the support layer water supply, so the latter limits the performance of the SISG evaporator. Thus, some restructuring of the support layer using commercial polymer foam has been performed. Wang et al. [30] solved this problem by adding highly absorbent polymer particles (SAP) to ordinary plastic floating foam. SAP (highly absorbent polymer) is a functional polymer material that becomes a hydrogel after absorbing water, and has good thermal insulation properties and abundant water channels. The support layer consists of cylindrical hollow foam with a diameter of 32 mm. SAP particles are added to the hollow foam. The experiment involved related tests on the non-woven hollow foam with 0.1 g of SAP particles added. The test results showed that, compared to the foam without SAP particles, under the same illumination intensity and at the same time, the temperature change in the foam with SAP particles added decreased slightly, mainly because the increased water evaporation resulted in some heat leaving the surface of the photothermal layer. This shows that the addition of SAP particles achieves a fast water supply.

#### 2.1.2. Functional Polymer Foam

The service environment of the evaporator affects its efficiency and service life in terms of various factors. For example, high salinity or pollution affect the durability of the support layer material [31]. Due to the limitations of traditional polymer foam in extreme environments, there is increasing research in developing functional composite foams. These include modified foams that are antibacterial, corrosion-resistant, etc. [32]. At the same time, recent studies have aimed to integrate functionalities into composite foams, combining capabilities for light and heat absorption with effective heat insulation [33,34]. Additionally, functional coating treatment and composite fabrication prior to the foaming process are also involved.

##### Coating Functionalized Polymer Foam

The high-performance and multi-functional coating material is configured and deposited on the surface of the polymer foam. The coated composite foam can be used as the support layer material to improve durability, functionality, etc. For surface coating treatment, one approach is to use a polymer component to coat the foam. This method usually improves the corrosion resistance and antibacterial properties of the foam [35]. Another method is adding functional nanomaterials to the surface coating to enhance its properties. It is noteworthy that, when employing surface-coating methods to treat foam, in addition to the usual polyurethane foam, melamine foam is also considered a good, original skeleton. This is because the open porous surface of melamine foam enables adhesion to any photothermal layer and is easy to combine with various coatings, without any surface degradation; compared with other polymer foams, melamine foam also has good thermal insulation properties and good pore orientation [36,37].

Pinto et al. [36] developed highly effective functional foams with antimicrobial properties (Figure 3a), explored the possibility of using them as antibacterial filters for water treatment, and decorated melamine foams with silver nanoparticles (ME/Ag). A uniform coating of silver nanoparticles (Ag NPs) with a diameter of less than 10 nm was formed in situ directly on the prop surface of the foam and then immersed in a AgNO_3_ solution. It was demonstrated that the nanoparticles adhered stably to the foam, and when filtered at flow rates of up to 100 mL/h·cm^2^, they were able to completely remove E. coli from the water, with the foam releasing less than 1 ppm of silver ions. After further dilution of the treated water, no bacterial regeneration was observed, and the level of mercury fell below the safety threshold for drinking water (0.1 ppm); subsequent combination with the photothermal layer can produce a multi-layer interfacial evaporator with an antibacterial support layer. Sometimes, it is necessary to ensure that the bottom support layer has a certain capacity to absorb heavy oil, under the premise of not destroying the photothermal layer, to achieve the effect of removing oil and ensuring interface evaporation at the photothermal layer. Traditional electric/solar heating absorption methods have low efficiency and high fire risk when heating high, easy fuel oil, which necessitates good flame retardancy in the support material. Shi et al. [38] developed a novel, highly efficient nanocoating for heavy oil absorption; the main component of the coating was rGO/Fe_3_O_4_, the coating showed both photothermal conversion ability and non-flammability, and it was loaded on the skeleton of melamine foam through a simple co-precipitation and dip-coating process. Thus, the modified foam could quickly and effectively remove high-viscosity heavy oil (3000 Pa·s). With the help of simulated sunlight, an absorption capacity for heavy oil of up to 75.1 g/g was achieved, and the absorption rate reached 9000 g m^−2^ min^−1^. At the same time, thanks to the dual photothermal conversion ability of functional rGO/Fe_3_O_4_ coating, the fabricated foam also has a certain photothermal evaporation ability.

##### Substrate Modified Functional Polymer Foam

After adding modified materials to the raw material to form a composite material, foaming treatment was performed, for example by adding nanoparticles to the base material. In this way, the raw material can enhance the structural strength and durability of the foam. Wang et al. [39] used this method to prepare nano-Ag-based photothermal foams with good light absorption and low thermal conductivity. The photothermal Ag NPs were added to the polyurethane base material and stirred evenly with an electric mixer, and then a certain amount of isocyanate was added. The resulting mixture was stirred at high speed, and then quickly poured into a mold and foamed at room temperature. Due to the introduction of nanosilver, the newly developed foam had good photothermal properties. Polyacrylamide absorbent material (PAM) was also added to the foam pores to provide a fast water supply channel for rapid water supply.

Hitherto, there has been much research on the preparation of commercial foam and modified foam as the support layers of a photothermal evaporator to take and optimize the advantages of polymer foam. However, at the same time, a polymer foam-based support layer still presents the disadvantages of complex processes, pollutant discharge, and difficulty in degradation, making it unfriendly to the environment.

### 2.2. Gel

#### 2.2.1. Hydrogel

A gel is a kind of flexible, solid polymer material with a unique three-dimensional structure, which is lightweight and floating, has good thermal conductivity, and is an ideal choice for support layers in a photothermal evaporator [40]. As common polymer gels, hydrogels are usually prepared with water as a dispersive medium and are crosslinked into a three-dimensional polymer network with acrylic polymer, acrylamide polymer, polyvinyl alcohol, etc. [41]. There are abundant hydrophilic groups such as hydroxyl, amino, and carboxyl groups in the polymer chains of the polymerized hydrogels. These functional groups are good at attracting water molecules into the three-dimensional networks of hydrogels, ensuring that they can quickly transport water to the photothermal layer when used as a support layer in a photothermal evaporator [21,40,42]. Excellent thermal conductivity can also prevent significant heat loss during light and heat evaporation. Evaporators using hydrogels as support layers have been used in water treatment fields such as seawater desalination and wastewater treatment.

Shi et al. [43] developed an interpenetrating network hydrogel evaporator with high mechanical strength, a high water evaporation rate, and good photoelectric conversion performance. Polyacrylamide (PAM) and crosslinking agents were used to form a three-dimensional crosslinked network structure, and the cis-hydroxyl group on the branch chain of galactomannan (GG) mannose formed a borate ester crosslinked network that enhanced the mechanical properties of the system. A photothermal layer composed of graphene oxide (GO) and Fe_3_O_4_ nanoparticles (Fe_3_O_4_ NPs) provides excellent photothermal transfer capability (Figure 4a). Under one sun irradiation, the GFCP evaporation rate reached 2.3 kg m^−2^ h^−1^, and the conversion efficiency was 90.7%. Wang et al. [44] also prepared a new type of reduced graphene oxide (rGO)/Cu_7_·_2_S_4_ polyvinyl alcohol (PVA)/polyacrylamide (PAM) composite hydrogel evaporator. The two-component pore structure of PVA/PAM hydrogel can improve the influence of water supply and drainage. The rGO/Cu_7_·_2_S_4_ provides photothermal conversion capability and achieved a high evaporation rate (2.19 kg m^−2^ h^−1^) (Figure 4b).

As an emerging green material, hydrogels show many unique advantages, excellent water absorption, heat insulation, biocompatibility, and flexibility, making hydrogels an ideal choice for the support layers of photothermal evaporators. However, hydrogels have limited durability, and prolonged exposure to sunlight can affect the evaporation rate. Therefore, when preparing hydrogels, consideration should be given to improving their durability. Some researchers are also trying to modify hydrogels’ properties further, such as their stain resistance [45,46], salt resistance [47,48], and bacteriostatic effects [49,50], to provide hydrogels with more prospective uses.

#### 2.2.2. Aerogel

Aerogels are materials that trap air molecules in porous (up to 99.8%) filled solids that prevent air from escaping. There are certain differences between aerogels and hydrogels in terms of the solvent selection, gelation method, special treatment, and drying method. Hydrogels are prepared by using water as the main solvent. Aerogels usually use organic solvents or other solvents [51]. Aerogel, a very low-density and high-surface-area porous material, also has a three-dimensional structure. There is much research on aerogels as supporting layers in photothermal evaporators. Usually, aerogels are combined with other functional photothermal materials to form multi-layer photothermal evaporators for interfacial evaporation. The raw materials for preparing aerogel are abundant, environmentally friendly, and cheap, and, most importantly, their biodegradability and biocompatibility are good, and they are easy to modify and functionalize [22,51,52].

Wu et al. [53] developed a multifunctional aerogel for interfacial evaporation. Plantago stalk (PC) was selected as a cellulose matrix and crosslinked with biomass hollow carbon tubes (hct) after chemical treatment to make an aerogel with a three-dimensional porous mesh scaffold (Figure 5a). The hct crosslinked with the aerogel has excellent photothermal conversion capability while retaining the characteristics of the aerogel as a flexible solid, which is porous and lightweight. The temperature of the optimized PC@HCT aerogel rapidly reaches 46.7 °C at the gas–liquid interface at 1.0 solar irradiance, and the evaporation rate is 1.86 kg m^−2^ h^−1^. In addition, PC@HCT aerogel has an obvious purification effect on seawater, heavy metal ion solutions, oil–water emulsions, and organic dye wastewater. It is important that the dry PC@HCT has good hardness and superelasticity in water. Furthermore, PC@HCT aerogels are green and recyclable, changing the healthy growing environment of sand and plants, thereby obtaining materials from nature and ultimately returning them to nature. Storer et al. [54] prepared a 3D photothermal aerogel composed of reduced graphene oxide (RGO) nanosheets, straw-derived cellulose fibers, and sodium alginate (SA) for solar steam generation. The photothermal aerogel shows an intense broadband light absorption of 96–97%. In the process of solar steam power generation, the 3D photothermal aerogel effectively reduces the radiation and convective energy loss while enhancing the environmental energy collection, resulting in a very high evaporation rate of 2.25 kg m^−2^ h^−1^ and an energy conversion efficiency of 88.9% under 1 solar irradiation (Figure 5b).

Aerogel is not, itself, inherently able to perform photothermal conversion. However, aerogel preparation is highly scalable and can be combined with different crosslinkers to meet different functional use scenarios. The aerogel’s high surface area and excellent thermal conductivity are also ideal for support layer materials in photothermal evaporators. However, the fragile characteristics of aerogel affect its durability to a certain extent, and steps to improve its durability and mechanical properties during preparation should be considered [55,56].

### 2.3. Biomaterials

#### 2.3.1. Natural Plants

Natural biomaterials have the inherent properties of hydrophilicity, cellulose components, and multi-dimensional holes and pores, making them a cost-effective source for support layers [57]. By now, a variety of plants, such as bamboo [58], pomelo peel [59], and wood [60] have been used directly or have inspired researchers to construct efficient SISG evaporators. Herein, wood has emerged as a favored candidate owing to its light weight and ability to float on the water’s surface. Moreover, it exhibits exceptional durability, retaining its structural stability after long periods of soaking. Furthermore, the inherent capillary structure of the wood facilitates gradual water transport to the photothermal layer, enhancing its suitability for applications in advanced SISG technology [61]. Extensive research on wood-based photothermal evaporators has been conducted, where unmodified wood has been directly used as a support layer (Figure 6a) [62,63]. Chen et al. [64] proposed a simple, cost-efficient, and scalable brushing method to prepare an aluminophosphate-treated wood (Wood@AlP) solar steam-generation device. The wood@AlP device can float on seawater and exhibits a high solar thermal efficiency of 90.8% with a net evaporation rate of 1.423 kg m^−2^ h^−1^ under one sun illumination (Figure 6b).

It is worth noting that the prolonged exposure of natural wood to high-salinity environments can result in the blockage of its capillary structure, thereby reducing overall water transmission. In view of the negative effect of salt crystallization on the water transport in the wood support layer, a series of wood-based evaporators with good salt drainage properties were investigated [66]. Shi et al. [67] explored a wood-based composite material composed of an evaporator with high evaporation performance and high salt resistance. They successfully obtained a solar-powered wood desalination device (PPy-E-Wood) through the in situ polymerization of pyrrole monomer on pretreated elastic wood. Efficient steam generation was successfully achieved through the synergistic effect of PPy NPs’ very light thermal conversion layer with a wood substrate with low thermal conductivity, micro-/nanopores, and channels. Bamboo, which has a natural porous capillary structure similar to wood, has also been selected for the support layer material in photothermal evaporators. Zhang et al. [58] developed a simple bamboo-based photothermal evaporator that uses a simple surface self-assembly technique to load polypyrrole (PPy) onto bamboo as a photothermal layer and bamboo as a support layer to make a solar evaporation device. The PPY-Bamboo solar evaporation device achieves 88% absorption of sunlight in the UV–vision-near-infrared region. The PPY-Bamboo solar evaporation device can achieve a high photothermal conversion efficiency of 76.87% under the light intensity of one sun.

In addition to biomaterials with natural porous structures, some biomaterials with hollow structures of beam tubes are also good choices for support layers. Fang et al. [65] took inspiration from rice straw and designed an evaporator that mimics the structure of rice straw as a support layer. Rice straw water transport is smooth, and the stratified channel of the stem has a spiral winding space and other multi-level structures, providing a variety of water channels, avoiding the scale, blockage, or scaling that may be caused by sand or soil particles, and maintaining its stable and anti-scaling capillary effect (Figure 6c). With the increasing innovations in natural plant-based support layer materials, diverse and unexpected naturally porous biomaterials, such as coconut shell [68], carrot [69], and loofah [70], can also be studied.

#### 2.3.2. Carbonized Biomaterials

The carbonization pre-treatment of natural green plants can enhance the material’s photothermal conversion ability. After carbonization pre-treatment, the biomaterial has the function of integrating the support layer and the photothermal layer, avoiding the need to re-design the photothermal layer. The biomass carbon prepared in this manner has a photothermal conversion ability, while retaining the supporting role of natural biological materials. Carbonized greens, such as carbonized woods and other plant-based carbon materials, are increasingly being explored for their effectiveness in interfacial water evaporation applications. These carbonized materials have shown promising properties, including high photothermal conversion efficiency due to their own dark aspect, good hydrophilicity inherited from proteins and celluloses, and scalability. Chen et al. [71] selected natural corn cob for a simple carbonization treatment (Figure 7a). Carbonized corn cob (C-corncobs) showed a rough and black surface, and maintained its natural porous structure. The surface carbonization layer acts as the light-absorbing layer. The “vesicle” structure of the central corn cob is similar to that of the foam composite material, and this special “ vesicle” structure facilitates thermal management and water supply, and provides sufficient steam channels for achieving light absorption and photothermal conversion. In addition to corn cobs, Xu et al. [72] found that mushrooms can also be used for solar evaporation. The natural porous capillary fiber structure of mushrooms can provide good water transport channels and inhibit heat diffusion. The subsequent carbonization treatment of natural mushrooms can further improve their evaporation efficiency. Zeng et al. [73] used carbonized waste durian skins to develop a new type of solar evaporator with a three-dimensional photothermal structure. The carbonized durian has a macroscopic three-dimensional pyramid, and a microscopic porous and petal-like structure that contributes to ideal light capture and absorption, and enables an extremely high absorption rate of up to 99% of the solar spectrum. The abundant porous microstructure inside the carbonized durian provides a good capillary effect for adequate water supply. There are many other green plants used in the preparation of biological carbon materials, such as oranges and daikon [74,75,76,77] (Figure 7b–d).

Studies are trying to improve the evaporation efficiency of carbonized biomaterials through simple modification while maintaining the advantages of easy degradation and low pollution. Therefore, modifying carbonized biomaterials further to improve their comprehensive evaporation performance or provide them with multiple functions has become a new research frontier.

#### 2.3.3. Modified Biomaterials

Modified biological materials are generally altered to enhance their performance as either the support layer or photothermal layer in interfacial water evaporators. However, basic biochar materials can serve as a support layer with a certain photothermal transfer ability. Under carbonization at high temperatures, natural structures such as vascular bundles in plants can be destroyed, affecting the water transport capacity inside the material. Solar interface evaporation often requires materials with strong water transfer capacities. In order to solve this problem, researchers have proposed incorporating modified materials that enhance water transport into carbonized biological materials. The resulting composite materials exhibit improved water transport performance. Guo et al. [78] employed loofah to prepare a MOF-801@CL composite material, which has good hydrophilicity and excellent water evaporation stability. Firstly, the loofah was selected as the substrate biomaterial. The loofah is easy to obtain and of low cost, and degrades easily without polluting the environment. By using MOF-801 as an adsorbent, a network structure similar to CL was synthesized using the in situ growth method, and MOF-801@CL was prepared. MOF-801 has good hydrophilicity and a rich pore structure, which can not only improve the hydrophilicity of CL but also provide additional water transport channels. Under the synergistic effect of the excellent water absorption of MOF-801, and the excellent photothermal conversion performance of CL, the water evaporation rate of MOF-801@CL is 1.42 kg m^−2^ h^−1^ under one sun, which is about 1.2 times that of CL, and the solar steam conversion efficiency is 88.9%. After the addition of MOF-801, the evaporation rate of CL was significantly increased.

In addition to the good water transport capacity required for materials used for interfacial water evaporation, some researchers have also loaded modified materials that can enhance the photothermal properties of the materials into the support layer substrate. Zhang et al. [79] selected sugarcane as the substrate due to its highly developed pore structure and excellent hydrophilicity. It is well known that conventional biomass carbon, including carbonized sugarcane, typically exhibits unsatisfactory absorption in the near-infrared region. Therefore, antimony-doped tin oxide (ATO) was introduced as a modified material to enhance the photothermal properties of biomaterials. ATO was loaded into sugarcane and then carbonized to obtain a composite solar evaporator CS@ATO with a solar energy absorption rate of 99%, a solar energy evaporation rate of 1.43 kg m^−2^ h^−1^, and an efficiency of 95.3%. In addition, the fabricated composite material was effective at treating wastewater, as experimental results showed that the concentration of heavy metal ions in wastewater was reduced by 3–4 orders of magnitude after CS@ATO treatment, with almost complete removal of dye from the wastewater.

By employing natural plants, carbonized biomaterials, and modified biological materials in the ways mentioned above, researchers aimed to create biomaterials that not only perform well in interfacial water evaporators but that also possess characteristics like biocompatibility, sustainability, and cost-effectiveness, which are advantageous for practical applications in water purification and desalination technologies.

To further clarify the comparison of the aforementioned support layer materials employed in the field of SISG evaporators, Table 1 was created to provide a brief summary of each type of material.

## 3. Support Layer Structure Optimization

In addition to appropriate material selection, optimizing the structural design of the support layer, which depends on various support layer materials, is crucial for improving the efficiency of the SISG evaporator. The optimization of interfacial evaporator structure addresses challenges, including heat loss and low evaporation efficiency. In addition to the traditional two-dimensional structure device, more efficient three-dimensional evaporator structures have been designed, such as conical structures [80], umbrella structures, and arch structures [81], and some emerging technologies, such as 3D printing technology, mainly based on polymers [82,83], are also being used in support layer structure design.

### 3.1. Cone Structure

The 3D cone structure is particularly advantageous due to its curved surface geometry, which facilitates multiple reflections and refractions of sunlight within the cone. This design enhances the probability of sunlight being captured and absorbed across a broad spectrum of wavelengths. From a macroscopic perspective, the cone structure promotes the efficient utilization of broadband light, thereby maximizing the photothermal conversion efficiency within the device [84]. Lv et al. [85] developed a 3D (3D) graded inverted conical solar evaporator consisting of a 3D copper foam skeleton cone and a graphene oxide functionalized micro-/nanostructure decorated on its surface by optimizing the foam structure of the support layer (Figure 8a). The high surface area of the skeleton structure provides sufficient space for the diffusion of steam from the boundary to the environment. The synergistic effect of the superhydrophilic skeleton and the air-planking paper wrapped around the skeleton prevents not only insufficient evaporation due to insufficient hydration but also heat loss due to the excessive evaporation of water. The rGo photothermal layer modified on its surface further improves the evaporation efficiency. Similarly, Xie et al. [86] developed a wood conical evaporator, which has multiple light reflections on the surface of the conical device, reducing diffuse reflection and promoting light absorption. When using this wood cone evaporator, the evaporation rate and efficiency reached 1.79 kg m^−2^ h^−1^. Bu et al. [87] designed a 3D carbon fiber cotton-based conical evaporator (CFC cone) with an adjustable water supply. The 3D CFC cone evaporator was designed to effectively collect incident sunlight, with an evaporation rate of 3.27 kg m^−2^ h^−1^ and a photothermal conversion efficiency of 194.4% under one sun. On the basis of the conical structure, further micropore structure optimization was carried out, and the bionic tree structure inspired by natural trees retained the advantages of the conical structure while achieving good co-ordination of fast and high-flux water transport and salt drainage [88].

### 3.2. Arch Structure

Coincidentally, the simple arch structure can also significantly enhance the overall evaporation performance. Xu et al. [89] prepared a three-dimensional arched solar interface evaporator through material selection and simple structural design (Figure 8b). The photothermal layer originally used for polymerized PPY on air cushion paper combines strong capillary action and photon capture capabilities, and the simple macro-support layer structure design (3D arch structure) can greatly reduce heat conduction losses, thereby increasing the surface evaporation temperature. Chen et al. [90] demonstrated a Janus arched solar evaporator with a hydrophobic top photothermal layer for effective light absorption, solar thermal conversion, and steam evaporation with an evaporation rate of 2.82 kg m^−2^ h^−1^ in pure water. The arched design of the evaporator makes full use of the upper space and effectively reduces the evaporator footprint. The ratio of evaporation area to occupied area is 1.57:1 (Figure 8c).

### 3.3. Other 3D Structures

Other innovative 3D structured SISG evaporators are also being developed. For instance, Wu et al. [91] developed a pyramid evaporator using 3D printing technology (Figure 8d). Surface-distributed micropores are formed on the prepared surface, endowing the pyramid evaporator with an ultra-fast water spreading property. Due to the designed morphology of the 3D structure with asymmetric grooves and the gradient microcavity arrays, the liquid film spreads on the structure surface displays a position-related liquid film thickness and temperature gradient along the sidewall, which further leads to the thermocapillary force inside the liquid film and the ability to capture energy from the surrounding environment to enhance water evaporation and energy efficiency. This design leverages the unique geometry of a pyramid to enhance solar energy absorption and heat retention, thereby improving the efficiency of water evaporation. Zhang et al. [92] proposed a three-dimensional (3D) cup-shaped evaporator based on carbonized sorghum straw (CSS). The 3D structure of this cup-shaped evaporator incorporates features that effectively absorb heat loss from reflected light and thermal radiation. The excellent light absorption, super hydrophile properties, and good thermal insulation of the CSS give the 3D cup evaporator an excellent evaporation rate (3.27 kg m^−2^ h^−1^) and energy efficiency (131.2%) under one sun irradiation. Three-dimensional cup evaporators also show excellent salt resistance and good acid and alkali resistance in the production of clean water from seawater and wastewater.

The trend towards 3D structure design represents the future of enhancing the evaporation rate and efficiency in interfacial evaporators. Compared to traditional two-dimensional designs, 3D solar-driven interface evaporators offer additional surface area under the same solar irradiation. This expanded surface area effectively accelerates the evaporation. Reasonable structural design and the synergistic effect of high-performance materials achieve high utilization efficiency in the evaporator. This approach not only enhances evaporation rates but also introduces innovative possibilities for advancing research in water treatment through interfacial evaporation technology [87,93,94,95,96].

## 4. Conclusions and Prospect

In this paper, the material selection and structural design of support layers in the field of multi-layer SISG progress were reviewed. Support layer material selection focuses on polymers and biomaterials. The polymers, which are mainly chosen as support layer materials, include foams and gels, while the biomaterials include natural plants, and carbonized and further modified biomaterials. Original unmodified materials like polymer foams and natural plants offer simplicity in preparation, and they often lack durability for practical applications, especially in harsh environments such as high-salinity conditions. Modified support layer materials have shown promise in enhancing performance to some extent, yet scaling them from laboratory settings to practical large-scale applications remains a challenge. In addition to the innovation of supporting layer material selection, recent research progress emphasizes the fact that structural optimization is also a key way to improve the efficiency of photothermal evaporators. Innovative 3D support layer structures, such as cone and arch designs, effectively regulate the light absorption area and steam flow resistance, presenting new avenues for surface structure engineering in enhancing photothermal evaporator performance.

The design of the support layer of the photothermal evaporator still faces many challenges. Compared with photothermal materials, the selection of support layer materials is relatively limited, and most studies focus on solid materials. The field of water treatment in the real world is much more complex. The support layer is not limited to the role of water transport support. The functionalization and integration of the support layer have become the focus of the subsequent design of photothermal evaporators. Broadening the selection of materials, improving the performance of the support layer, and making it commercially applicable will need to be considered in future research. It is believed that SISG technology, supported by various advanced support layers, can help drive eco-friendly, cost-effective, and sustainable development in the field of water purification.

## Figures and Tables

**Figure 1 polymers-16-02427-f001:**
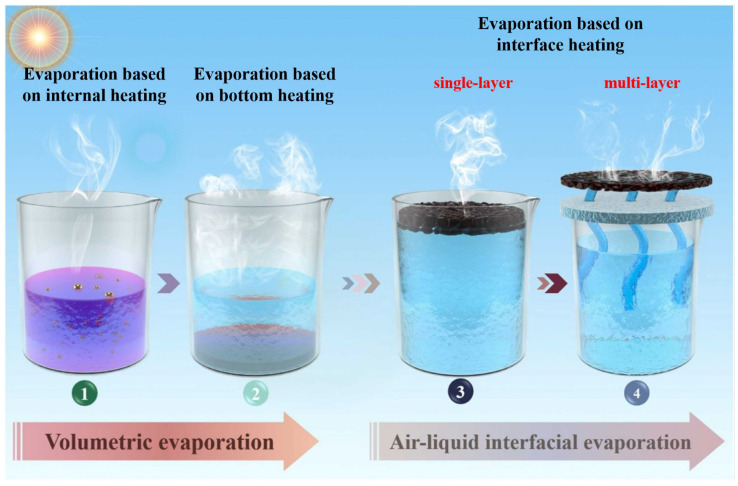
The overall material/structural scheme of an interfacial solar evaporation system [11].

**Figure 2 polymers-16-02427-f002:**
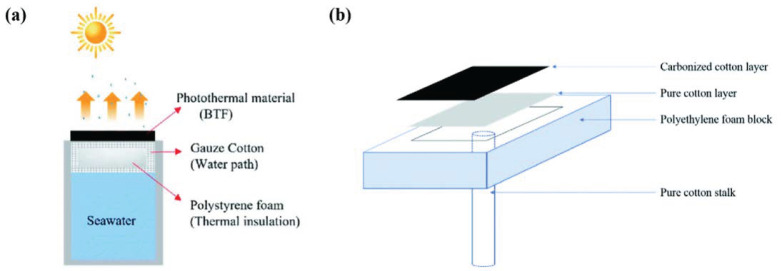
The schematic of the seawater evaporation experiment designed by Nguyen (**a**) [28] and schematic representation of the CC solar steam-generation device designed by Kiriarachchi (**b**) [29].

**Figure 3 polymers-16-02427-f003:**
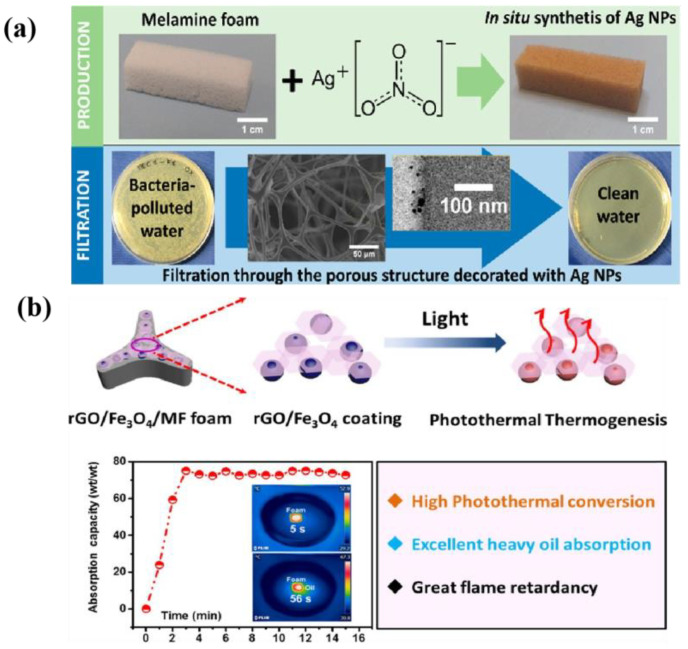
Photographs of the treated ME/Ag foams (**a**) [36], and schematic illustration for the preparation of the rGO/Fe_3_O_4_/MF foam (**b**) [38].

**Figure 4 polymers-16-02427-f004:**
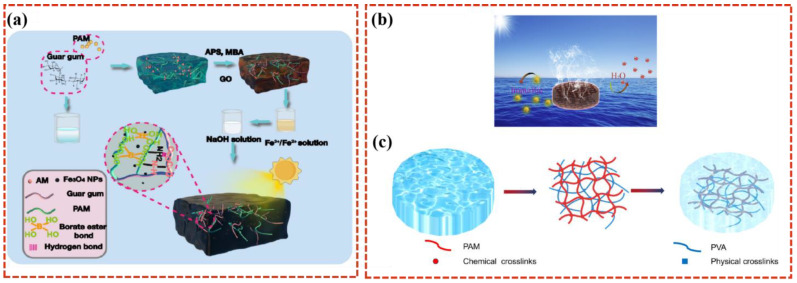
Flowchart of GFGP hydrogel evaporator preparation (**a**) [43]. A schematic diagram of the hydrogel mixing device generated through solar steam generation (**b**). A schematic diagram of the PVA/PAM hydrogel polymer network’s formation (**c**) [44].

**Figure 5 polymers-16-02427-f005:**
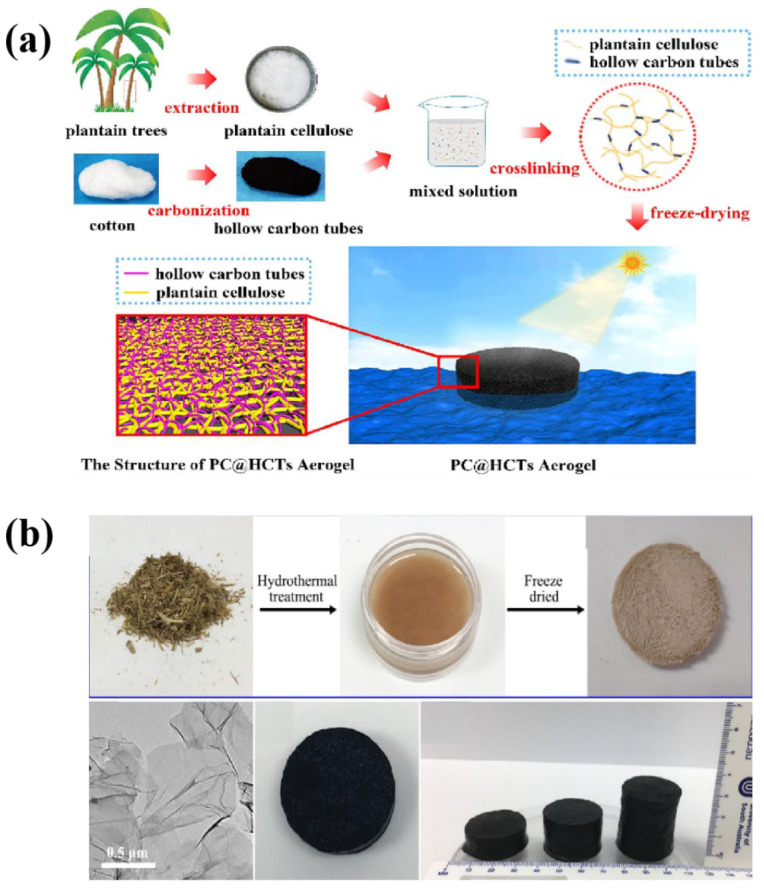
Schematic of the fabrication of the solar evaporator with the PC@HCTs aerogel (**a**) [53], and schematic of the fabrication of the solar evaporator with the RGO–SA–cellulose aerogel (**b**) [54].

**Figure 6 polymers-16-02427-f006:**
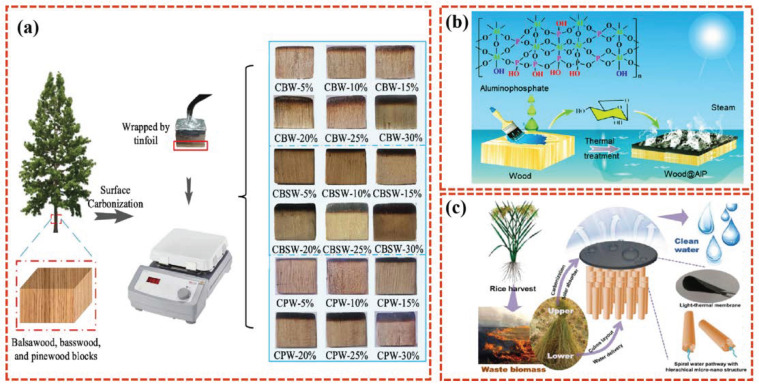
The fabrication of a carbonized wood-based solar evaporator through surface carbonization (**a**) [62]. Schematic image showing the fabrication of the Wood@AlP-based solar steam-generation device (**b**) [64]. Illustration of the design route of the biomass-based solar steam-generation device (**c**) [65].

**Figure 7 polymers-16-02427-f007:**
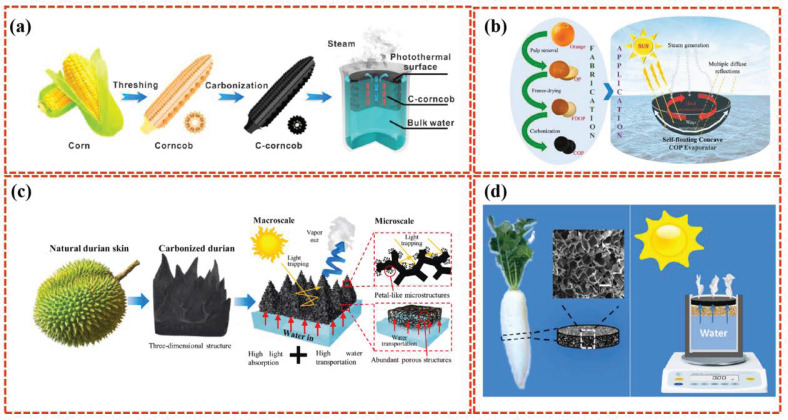
Schematic of the C-corncob-derived SISG device (**a**) [71]. Schematic of the 3D carbonized orange peel device (**b**) [75]. Schematic of a carbonized durian solar evaporator. Excellent solar light absorption is maintained by the light trapping of three-dimensional pyramid porous structures together with petal-like microstructures (**c**) [73]. Schematic of the carbonized daikon device (**d**) [77].

**Figure 8 polymers-16-02427-f008:**
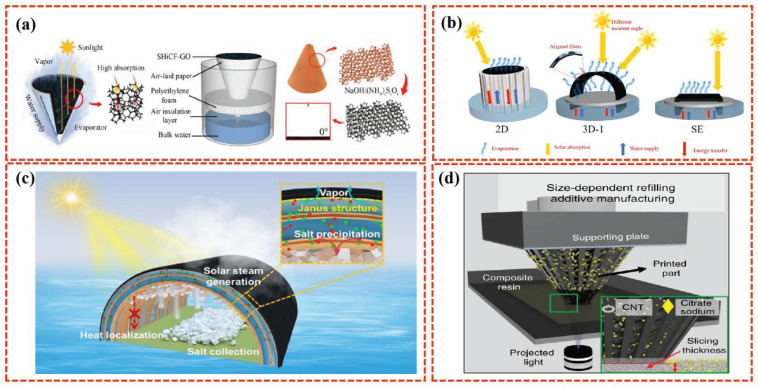
Schematics of the proposed light absorption principle, evaporator, fabrication process, and characterization of the 3D inverted conical solar evaporator (**a**) [85]. Some schematic diagrams of the arch structure evaporation device (**b,c**) [89,90]. Schematic configuration of size-dependent resin refilling induced additive manufacturing based on the continuous DLP 3D printing system (**d**) [91].

**Table 1 polymers-16-02427-t001:** The brief review of different types of support layer material in multi-layer SISG evaporators.

Support Layer Category	Typical Materials	Main Advantages
Commercial polymer foam	Polyurethane (PU) foams Polystyrene (PS) foams Polydimethylsiloxane (PDMS) foam	Light weight, self-floating, low thermal, conductivity.
Functional polymer foam	Coating Functionalized Polymer Foam Substrate Modified Functional Polymer Foam	Light weight, self-floating, high-performance, multi-functionalization.
Gels	Hydrogel Aerogel	Environmentally friendly, unique transporting water structure.
Biomaterials	Natural plants Carbonized biomaterials Modified biomaterials	Cost-effective, rich source environmentally friendly.

## Data Availability

Not applicable.

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
