# Peer review of "Development Status of Solar-Driven Interfacial Steam Generation Support Layer Based on Polymers and Biomaterials: A Review"

_polymers, 2024, doi:10.3390/polym16172427_

Round 1
Reviewer 1 Report
Development Status of SISG Support Layer Based on Polymers and Biomaterials
The reviewed work discusses the use of various materials to create a supporting layer in solar-driven interfacial steam generation (SISG). Although the word Polymer appears in the title, the work does not consider methods of creating polymers intended for this application or their specific properties. Looking at the entire manuscript, the appropriate place for publication - after removing the errors, will be MDPI Energies.
The work is written with numerous grammatical errors and incorrect selection of words and concepts. Due to these errors it cannot be published. The work requires proofreading by a native speaker who also knows the issues discussed.
The work should include a more detailed discussion of the principles of operation of the entire SIGS system, not just the support layer.
Below is a list of wording errors encountered in the first few pages of the manuscript.
Line 34 - improper use of "etc." - "etc." indicates that the list is incomplete and that there are additional items that the reader can infer or that are obvious in the context, which is not obvious in this case.
Line 36 – “evaporation of water to generate steam” – the word steam means water vapor under pressure and usually at a temperature above 100°C. The examples in the illustrations show that water evaporates into the air and can be called "vapour".
Line 36-37 – “The condensed water collected afterward effectively desalinates seawater.” The sentence is illogical and ungrammatical. “condensed water…desalinates seawater.”
Line 39 – “In commonly” replace “Generally”.
Line 51 – replace “efficiency” with “efficient”.
Line 57 – “often leads” replace “often leads to”
Figure 1 “Evaporation based on integral heating” - maybe “internal heating”
Line 64 – “photothermal layer in evaporation” change to “Photothermal layer used in evaporation”.
Line 65-67 – “For example, after years of research, it has been found that carbon-based materials [16], metal materials [17, 18] and semiconductor materials have emerged as optimal choices for photothermal layer materials. ” The sentence is illogical from the point of view of mathematics - replace the word "optimal" with "good".
Line 83 – “This work, through current work, aims to provide” - what does that mean?
Line 95 – “light buoyancy” – remove “light”
Line 96 – change "often exhibiting" to "which often exhibit"
Line 103-105 – change to “Moreover, reasonable treatments of biomaterials, including drying, carbonization, and modification with functional components, enable them to achieve high evaporation efficiency and suitability for diverse practical applications.”
Line 112 – change “float” to “structure”
Line 115 – "insulates against heat after" change to "insulates against heat loss"
Line 123-124 “provides a larger surface area, enabling more water absorption and further improving evaporation efficiency.” - In this application water absorption is not necessary, but rather diffusion and capillary water flow.
Line 130 - 131 - the authors introduce the concept of "evaporation efficiency 83%" without defining it before.
In many places, authors provide the first, middle and third names and surname of the first author (Thi Kieu Trang Nguyen et al.), while the common practice is to provide only the surname. By the way, et al. It should be written in italics because it comes from Latin
The caption to Figure 2 should be corrected.
Line 147 – replace "speed" with "rate"
Line 150 – “Wang et al. [30] added this problem by adding” – maybe “solved this problem”
Line 153-155 - “The supporting layer is made of cylindrical hollow foam with a diameter of 32 mm. SAP particles are added into the hollow foam.” – does this mean that the foam was made into a pipe and filled with SAP particles?
Line 160-161 – please correct this clumsy wording
Line 170 – what does the word “pre” mean here?
Similar word choice errors are found throughout the article.
The work cannot be published in this condition.
Comments on the Quality of English LanguageEnglish is clumsy.
Author Response
General comment: The reviewed work discusses the use of various materials to create a supporting layer in solar-driven interfacial steam generation (SISG). Although the word Polymer appears in the title, the work does not consider methods of creating polymers intended for this application or their specific properties. Looking at the entire manuscript, the appropriate place for publication - after removing the errors, will be MDPI Energies.
Response: Thanks very much for your comment. We apologize for the confusion caused. Sincerely, we considered that the polymers are commonly utilized as support layers in SISG development. Section 2.1-2.2 of the manuscript extensively introduce the advantages and current trends pertaining to polymer as the support layer material. Additionally, a briefly overview of preparation methods of polymer-based support layer materials, such as the utilization of gel as a supporting layer material, is provided. The progress about polymers applied in support layer takes up a large part of this manuscript. For instance, we further modified some part to emphasize the importance of polymer in this field. Therefore, we consider that the modified manuscript is also suitable for the journal of MDPI Polymers, offering valuable references for researchers in the field of polymer. Thanks very much for your suggestion and understanding.
The mainly modified parts are listed as below:
Lines 19-24 in Abstract section: This review focuses on the support layer, which is relatively neglected in evaporator development. It summarizes existing progress in the field of multi-layer interface evaporators, based on various of polymers and biomaterials, along with their advantages and disadvantages. Specifically, polymer-based support layers are mainly reviewed, including polymer foams, gels, and their corresponding functional materials. While biomaterial support layers, including natural plants, carbonized biomaterials, and other innovation biomaterials.
Lines 103-104: At present, most common support layer materials are based on polymers.
Lines 483-487: In addition to the traditional two-dimensional structure device, more efficient three-dimensional evaporator structures have been designed, such as conical structures [80], umbrella structures, and arch structures [81], and some emerging technologies, such as 3D printing technology mainly based on polymers [82, 83], are also being used in support layer structure design.
Lines 556-560: In this paper, material selection and structural design of support layer in the field of multi-layer SISG progress are reviewed. Support layer material selection focuses on polymers and biomaterials. The polymers, which are mainly chosen as support layer materials, include foams and gels, while the biomaterials include natural plants, carbonized, and further modified biomaterials.
Lines 463-465. Additional summarized table to overview the different materials chosen for support layer in the field of SISG evaporators researches further support the important of polymers utilization.
Table 1. The brief review of different types of support layer material in multi-layer SISG evaporators.
Support layer category |
Typical materials |
Main advantages |
Commercial Polymer Foam |
Polyurethane (PU) foams Polystyrene (PS) foams Polydimethylsiloxane (PDMS) foam |
Light weight, self-floating, low thermal, conductivity. |
Functional Polymer Foam |
Coating Functionalized Polymer Foam Substrate Modified Functional Polymer Foam |
Light weight, self-floating, high-performance, multi-functionalization. |
Gels |
Hydrogel Aerogel |
Environmentally friendly, unique transporting water structure. |
Biomaterials |
Natural Plants Carbonized Biomaterials Modified Biomaterials |
Cost-effective, rich source environmentally friendly. |
Comments 1: The work is written with numerous grammatical errors and incorrect selection of words and concepts. Due to these errors it cannot be published. The work requires proofreading by a native speaker who also knows the issues discussed.
Response 1: Thanks very much for your comment. We are sorry for our careless mistakes about the language and writing style. We have received assistance from a professional group, the MDPI Editorial Team, and have carefully checked the manuscript to polish our revised version. All the errors you mentioned in your revision comments have been corrected, and we have also carefully checked and corrected the grammatical spelling of the remaining parts
The manuscript has been carefully revised to minimized formatting issues and grammatical/typo errors. All the corrections were highlighted in Red for your reviewing in revised manuscript. In addition, the certification of English Editing, which got from the MDPI Editorial Team, has also been attached.
The modified errors are follows:
Line 24, “In addition” to “Additionally”
Line 25, “are” to “were”
Line 28, “to” to “for”
Line 30, “for the effectively respond” to “to enable effective responses”
Line 36, insert “and”
Line 37, delay“etc.”
Line 38, “the evaporation of water to generate steam” to “vapour”
Lines 38-39, “The condensed water collected afterward effectively desalinates seawater.” to “Collect the condensed water after evaporation to complete the desalination of sea water.”
Line 41, “In commonly” to “Generally”, “to” to “for”, “unitize” to “unitizing”
Line 42, “In commonly” to “Generally”
Lines 51-53, “Advanced fabrication of photothermal evaporator, including reasonable materials employment and optimized structure design, is the key to realize efficiency interfacial water evaporation.” to “The advanced fabrication of photothermal evaporators, including the employment of reasonable materials and optimized structure design, is the key to realizing efficient interfacial water evaporation.”
Line 54, “evaporator” to “evaporators”, “divided” to “classified”
Line 55, “evaporator” to “evaporators”
Line 56, insert “directly”
Line 60, insert “to”
Line 61, “Overall” to “overall”, “Material” to “material”, “Structural” to “structural”, “Scheme” to “scheme”, “Interfacial” to “interfacial”, “Solar” to “solar”, “Evaporation” to “evaporation”, “System” to “system”
Line 61, Figure 1, “integral heating” to “internal heating”
Line 81, “optimal” to “good”
Line 83, “of photothermal layer” to “on the support layer”, “the improving of the supporting layer” to “improving the support layer”
Line 85, insert “an”
Line 90, “insert “a”
Line 91, “investigates in one place” to “explores”
Line 94, insert “and”, delay “through current work”, “provide” to “describe”
Line 95, insert “the”
Line 100, “typically” to “typical”
Lines 102-103, delay “so the”, insert “Thus, the thermal conductivity and water absorption characteristics should be carefully considered when choosing the support layer material.”
Line 105, “polymer based” to “polymer-based”, “featured” to “characterized”
Line 107 “light buoyancy” delay “light”
Line 108, delay “Besides”, “aerogels” to “Aerogels”, insert “and they”
Line 109, “by” to “because of”
Line 111, “by” to “using”, “hold” to “possesses”, insert “a”
Line 112, “eco-friendly” to “eco-friendliness”
Line 114, “structure” to “structures”
Line 115, “on” to “of”, “modified by” to “and modification with”
Line 116, delay “and etc.”, “which make” to “enable”, insert “to”, “efficiency of evaporation” to “evaporation efficiency”
Line 117, “suitable” to “suitability”
Line 124, “well as” to “their”, “cost-effective” to “cost-effectiveness”
Line 125, “float” to “structure”
Line 108, “often exhibiting” to “which often exhibit”
Line 128, “after” to “loss”
Lines 129-130, “This insulation does the effective function to maintain high evaporation efficiency in the evaporator.” to “This insulation is responsible for effectively maintaining high evaporation efficiency in the evaporator.”
Lines 131-132, “In addition, the other important feature, porous structure, of commercial plastic foams further offers several advantages during application in SISG technology.” to “The porous structure of commercial plastic foams is another important feature that confers several advantages during application in SISG technology.”
Lines 135-136, “enabling more water absorption and further improving evaporation efficiency.” to “allowing more water to absorb and diffuse into capillary water flow further improving evaporation efficiency.”
Line 137, insert “it”
Line 140, “purposefully” to “effectively”
Line 142, “reached on” to “showed”, insert “a good”
Line 143-144, delay “and 83 %, respectively.”
Line 146, “Thi Kieu Trang Nguyen” to “Thi”
Line 150, “Besides” to “Additionally”, “Hiran D. Kiriarachchi” to “Hiran”
Line 152, “avoiding” to “preventing”
Line 153, “evaporating” to “evaporation”
Line 155, “Schematic” to “schematic”, “Thi Kieu Trang Nguyen” to “Thi”
Line 156, “Schematic” to “schematic”, “Hiran D. Kiriarachchi” to “Hiran”
Line 157, delay “However”, “due” to “Due”, “accelerated by a lot of” to “achieved in many recent”
Line 158, “been already greater than” to “long surpassed”, “speed” to “rate”
Line 159, insert “the”
Line 160, “have been done” to “has been performed”
Line 165, “carried out” to “involved”
Line 166, insert “of”
Line 165-166 “The support layer consists of cylindrical hollow foam with a diameter of 32 mm. SAP particles are added to the hollow foam.” We have consulted the original literature, this mean that the foam was made into a pipe and filled with SAP particles.
Line 168, “of” to “in”
Line 170, “took away some heat from” to “resulted in some heat leaving”, “it” to “This”
Lines 171-172, delay “and the evaporation of evaporator water increases linearly.”
Line 174, “by” to “in terms of”, “polluted environments” to “pollution affect”
Line 177, “corrosion resistance” to “corrosion-resistant”
Line 178, “aim” to “have aimed”
Line 180, “Besides” to “Additionally”, “pre to foam” to “prior to the foaming”
Line 183, “high performance” to “high-performance”, “are” to “is”
Line 185, “have better” to “improve”
Line 187, “allows the foam to have better” to “usually improves the”, insert “of the foam”
Line 190, “also selected as an well” to “is also considered a good”
Line 192, insert “and is”
Line 199, “is” to “was”
Line 200, “the” to “a”
Line 201, “adhere” to “adhered”
Line 204, “reaching below the safe” to “and the level of mercury fell below the safety”
Line 205, “mercury” to “water”
Line 206, insert “an”
Line 207, “absorption for” to “capacity to absorb”
Line 211, “requires” to “necessitates”
Line 217, “the” to “an”, “of: to “for”, “is” to “of”
Line 218, insert “was achieved”, “is up to” to “reached”
Line 226, “By” to “In”
Line 228, “nano Ag-based” to “nano-Ag-based”
Line 230, insert “and”
Line 231, “is” to “was”
Line 232, insert “and”
Line 233, “has” to “had”
Line 236, “have” to “has”, “many researches” to “much research”
Line 238, “But” to “However”, insert “a”
Line 239, “facing” to “presents”, “process” to “processes”
Line 240, “as well as difficult to degrade which is unfriendly to environment” to “and difficulty in degradation, making it unfriendly to the environment”
Line 243, “Gel” to “a gel”, insert “a”
Line 245, “supporting layer” to “support layers”
Line 246, “gel” to “gels”
Line 247, “by” to “with”
Line 251, “it” to “they”, insert “quickly”, delay “timely”
Line 253, “a lot of” to “significant”
Line 254, “when” to “during”, “based on” to “using”
Line 258, insert “a”, insert “good”
Line 259, “agent” to “agents”
Line 262, “to enhance” to “that enhanced”
Line 265, “reaches up to” to “reached”, “is” to “was”, insert “the”
Line 276, “research” to “researchers”, “hydrogels” to “hydrogels’ properties”
Line 277, insert “their”, “bacteriostasis” to “and bacteriostatic effects”
Line 278, “which make hydrogels have a wider use prospect.” to “to provide hydrogels with more prospective uses.”
Line 280, “The” to “A”
Line 281, “by” to “through”, “The” to “A”
Line 282, “network” to “network’s”
Line 286, insert “the”
Line 288, “low density” to “low-density”
Line 289, “high surface area” to “high-surface-area”
Line 290, “researches” to “research”
Line 293, insert “and”
Line 304, insert “an”
Line 306, “What's more” to “Furthermore”
Line 308, insert “them”
Line 310, “straw derived” to “straw-derived”
Line 317, “itself does not” to “is not”, “possess the ability of” to “able to perform”
Line 318, “has high scalability” to “is highly scalable”
Lines 322-323, “it should be considered to improve its durability and mechanical properties during preparation.” to “steps to improve its durability and mechanical properties during preparation should be considered.”
Line 326, “Schematic” to “schematic”
Line 330, “with” to “have the”
Line 331, insert “and”
Line 332, insert “a”
Line 335, insert “the”
Line 336, “time” to “periods of”
Line 340, insert “has been conducted”, “as support layer directly” to “has been directly used as a support layer”
Line 343, “exhibit” to “exhibits”
Line 347, insert “a”, “by” to “through”
Line 351, insert “the”, “in high salinity” to “to high-salinity”
Line 354, “of” to “in the”, “wood based” to “wood-based”
Line 358, “by in-situ” to “through the”
Line 359, “is” to “was”
Line 360, “high-light” to “very light”
Line 377, insert “the”, “on nature plant based” to “in natural plant-based”
Line 379, insert “and”, delay “etc.”, “used in the study” to “studied”
Line 381, insert “The”, “make the material have enhanced” to “enhance the material’s”
Lines 383-384, “so as to avoid re-designing” to “avoiding the need to re-design”
Lines 384-385, “by this operation have the ability of photothermal conversion” to “in this manner has a photothermal conversion ability”
Line 386, “like” to “such as”
Line 390, “as well as” to “and”
Line 394, “light absorbing” to “light-absorbing”
Line 400, “Subsequent” to “The subsequent carbonization”
Line 401, “continue to improve the” to “further improve their”, insert “used”
Line 405, “provides” to “enables”
Line 408, insert “and”, delay “so on”
Line 417, “by” to “through”
Line 418, “small” to “low”
Line 419, “make” to “provide”, “multi-functions” to “multiple functions”
Line 420, “edge” to “frontier”
Line 425, “Due to” to “Under”
Line 426, “will be” to “can be”, delay “to a certain extent”, “Thereby, water transport capacity inside the material was affected.” to “affecting the water transport capacity inside the material.”
Line 428, “capacity” to “capacities”
Line 433, insert “and”
Line 434, “easy to degrade without pollution to the environment” to “degrades easily without polluting the environment”, insert “an”
Line 435, “by in-situ” to “using the in situ”
Line 436, insert “a”
Line 448, “exhibit” to “exhibits”
Line 451, “is” to “was”
Line 454, “experiment” to “experimental”
Line 458, “as well as” to “and”
Line 459, “in above ways” to “in the ways mentioned above”
Line 480, insert “which”, “plays a crucial role in” to “is crucial for”
Line 481, insert “the”
Line 484, “are” to “have been”, “for example” to “such as”
Line 485, “structure” to “structures”, insert “and”
Line 486, insert “mainly based on polymers”, insert “being”
Line 491, “possibility” to “probability of”, “will be” to “being”
Line 493, insert “the”
Line 500, “air planking” to “air-planking”
Line 505, “By” to “When”
Line 506, “are up to” to “reached”
Line 507, “cotton based” to “cotton-based”
Line 508, “Cone” to “cone”, “is” to “was”
Line 548, “superhydrophile” to “super hydrophile”
Line 551, “3D” to “Three-dimensional”
Line 552, “process of producing” to “production of”
Line 558, “of” to “in the”
Line 566 “including” to “include”
Line 571, “yet their scalability” to “yet scaling them”
Line 589, “improve” to “improving”, “make” to “making”
Comments 2: The work should include a more detailed discussion of the principles of operation of the entire SIGS system, not just the support layer.
Response 2: Thank you for pointing this out. We agree with this comment. Therefore, we have modified and supplemented this article with discussion of the principles of operation of the entire SIGS system in Introduction sections.
The modified part is as follows:
Lines 65-79: The support layer is usually located at the bottom, providing stable structural support and preventing direct contact between bulk water and the thermal layer, reducing heat loss. It is typically constructed with lightweight materials that can float on the water. The working principle of the multi-layer evaporator is that the top photothermal layer absorbs the incident sunlight and converts it into heat energy. Through capillary action, water gradually infiltrates the photothermal layer from bottom to top through the water vapor transport channel arranged in the support layer under the photothermal layer and accumulates into a thin surface water layer at the top of the photothermal layer. The photothermal layer heats the water in the surface water layer, causing it to evaporate into steam. The water in the surface water layer gradually evaporates, and the impurities in the water are retained in the photothermal absorption layer because they cannot evaporate, thus achieving desalination. Multi-layer interfacial evaporators for efficient interfacial evaporation technology are attracting increased attention from researchers for the scientifically based optimization of their parameters such as nature, morphology, thickness, interface, and structure.
Thanks again for your carefully review and valuable comments, which improved the quantity of our manuscript highly! Sincerely thanks!

Reviewer 2 Report
Comments and Suggestions for Authors
Haipeng Yan et al. presented a review article titled "Development Status of SISG Support Layer Based on Polymers and Biomaterials." The article makes a commendable effort to provide information on support layer materials in multi-layer interfacial water evaporators, structure design strategies of support layer, the sustainability and effectiveness of interfacial evaporation technology supported by citations from recent literature.
However, some of the literature and findings presented lack sufficient mechanisms and summary tables, making them difficult for readers to fully understand. Therefore, in its current form, the manuscript may not be suitable for publication in our journal. I recommend that the authors revise the manuscript to address these issues.
Comments:
1- A summary table representing different layered interfacial water evaporators, their use/effectiveness, energy consumption, mode of action, and comparison in terms of efficiency, cost, etc. is necessary for comparative claims of the said technology.
2- Given that the authors are affiliated with mechanical engineering, automation, chemistry, and environment, it is recommended to include a chemical and environmental aspect should be included. Additionally, providing graphical representations would greatly aid young researchers working in this field.
3- Figures presented in the review require permission to avoid copyright issues. However, for Figure 1 if drawn by authors then it is still unclear and required to fill in the detailed working mechanism.
4- Review objectives are vague and confusing and English is written poorly. Especially, on page 3, lines 79-83 don’t give clear meaning.
5- Usually Abstract represents what you have done/written in the article and the conclusion depicts your findings/what you have achieved after doing so much literature on said topic. In this perspective, I would suggest authors to rewrite the abstract and conclusions sections.
6- Please arrange all the references into a single format. e.g. journal name in full form or abbreviated form. E.g., References 30 and 36 are from the same journal but the names are in a different format. In reference 39 abbreviation is used for the journal name, please check all the references in this context.
Comments on the Quality of English LanguageComments:
1- A summary table representing different layered interfacial water evaporators, their use/effectiveness, energy consumption, mode of action, and comparison in terms of efficiency, cost, etc. is necessary for comparative claims of the said technology.
2- Given that the authors are affiliated with mechanical engineering, automation, chemistry, and environment, it is recommended to include a chemical and environmental aspect should be included. Additionally, providing graphical representations would greatly aid young researchers working in this field.
3- Figures presented in the review require permission to avoid copyright issues. However, for Figure 1 if drawn by authors then it is still unclear and required to fill in the detailed working mechanism.
4- Review objectives are vague and confusing and English is written poorly. Especially, on page 3, lines 79-83 don’t give clear meaning.
5- Usually Abstract represents what you have done/written in the article and the conclusion depicts your findings/what you have achieved after doing so much literature on said topic. In this perspective, I would suggest authors to rewrite the abstract and conclusions sections.
6- Please arrange all the references into a single format. e.g. journal name in full form or abbreviated form. E.g., References 30 and 36 are from the same journal but the names are in a different format. In reference 39 abbreviation is used for the journal name, please check all the references in this context.
Author Response
General comment: Haipeng Yan et al. presented a review article titled "Development Status of SISG Support Layer Based on Polymers and Biomaterials." The article makes a commendable effort to provide information on support layer materials in multi-layer interfacial water evaporators, structure design strategies of support layer, the sustainability and effectiveness of interfacial evaporation technology supported by citations from recent literature.
However, some of the literature and findings presented lack sufficient mechanisms and summary tables, making them difficult for readers to fully understand. Therefore, in its current form, the manuscript may not be suitable for publication in our journal. I recommend that the authors revise the manuscript to address these issues.
Comments 1: A summary table representing different layered interfacial water evaporators, their use/effectiveness, energy consumption, mode of action, and comparison in terms of efficiency, cost, etc. is necessary for comparative claims of the said technology.
Response 1: Thank you for pointing this out. We have briefly made a summary table of different types of support layers materials in multi-layered interfacial evaporators, including the typical materials and main advantages. The table was named Table 1 and present it on page 3.
The table is as follows:
Table 1. The brief review of different types of support layer material in multi-layer SISG evaporators.
Support layer category |
Typical materials |
Main advantages |
Commercial polymer foam |
Polyurethane (PU) foams Polystyrene (PS) foams Polydimethylsiloxane (PDMS) foam |
Light weight, self-floating, low thermal, conductivity. |
Functional polymer foam |
Coating Functionalized Polymer Foam Substrate Modified Functional Polymer Foam |
Light weight, self-floating, high-performance, multi-functionalization. |
Gels |
Hydrogel Aerogel |
Environmentally friendly, unique transporting water structure. |
Biomaterials |
Natural plants Carbonized biomaterials Modified biomaterials |
Cost-effective, rich source environmentally friendly. |
In addition, as you mentioned in the review comments to make a summary table to compare the relevant characteristics of different layer of evaporators. However, we divide the evaporator types into single-layer and multi-layer evaporators in the review, and mention that single-layer evaporators do not have a separate support layer, which makes it hard to reviewed in briefly. Based on the main review of the different type of support layer materials, we create Table 1 to mainly summarized and compared the support layer materials. This may much suitable for our manuscript. We sincerely thank you for your valuable advice.
Comments 2: Given that the authors are affiliated with mechanical engineering, automation, chemistry, and environment, it is recommended to include a chemical and environmental aspect should be included. Additionally, providing graphical representations would greatly aid young researchers working in this field.
Response 2: Thank you for pointing this out. As you said, chemical and environmental components should be included in the review process. We have mentioned this in some places in the article. For example, when introducing the support layer material of gels, we cited some chemical knowledge.
The modified part is as follows:
Lines 243-256: “A gel is a kind of flexible solid polymer material with a unique three-dimensional structure, which is lightweight and floating, has good thermal conductivity, and is an ideal choice for support layers in a photothermal evaporator [40]. As common polymer gels, hydrogels are usually prepared with water as a dispersive medium and are crosslinked into a three-dimensional polymer network with acrylic polymer, acrylamide polymer, polyvinyl alcohol, etc. [41]. There are abundant hydrophilic groups such as hydroxyl, amino, and carboxyl groups in the polymer chains of the polymerized hydrogels. These functional groups are good at attracting water molecules into the three-dimensional networks of hydrogels, ensuring that they can quickly transport water to the photothermal layer when used as a support layer in a photothermal evaporator [21, 40, 42]. Excellent thermal conductivity can also prevent significant heat loss during light and heat evaporation. Evaporators using hydrogels as support layers have been used in water treatment fields such as seawater desalination and wastewater treatment.”
In terms of environment, we mainly reviewed the environmental friendliness of the support layer material, which was mainly reflected in section 2.2-2.3 of the manuscript.
Lines 110-112: At the same time, a support layer fabricated using biomaterials possesses the advantages of a wide range of sources, low cost, and eco-friendliness [23].
Lines 292-295: The raw materials for preparing aerogel are abundant, environmentally friendly, and cheap, and, most importantly, their biodegradability and biocompatibility are good, and they are easy to modify and functionalize [22, 51, 52].
Lines 458-462: By employing natural plants, carbonized biomaterials, and modified biological materials in the ways mentioned above, researchers aim to create biomaterials that not only perform well in interfacial water evaporators but also possess characteristics like biocompatibility, sustainability, and cost-effectiveness, which are advantageous for practical applications in water purification and desalination technologies.
Comments 3: Figures presented in the review require permission to avoid copyright issues. However, for Figure 1 if drawn by authors then it is still unclear and required to fill in the detailed working mechanism.
Response 3: Thank you for pointing this out. All the permissions of the presented figures have been required and submitted journal of MDPI Polymer. In addition, we have enhanced the explanation of Figure 1 in more detail, in lines 65-79, and also presented as follow:
“The support layer is usually located at the bottom, providing stable structural support and preventing direct contact between bulk water and the thermal layer, reducing heat loss. It is typically constructed with lightweight materials that can float on the water [15]. The working principle of the multi-layer evaporator is that the top photothermal layer absorbs the incident sunlight and converts it into heat energy. Through capillary action, water gradually infiltrates the photothermal layer from bottom to top through the water vapor transport channel arranged in the support layer under the photothermal layer and accumulates into a thin surface water layer at the top of the photothermal layer. The photothermal layer heats the water in the surface water layer, causing it to evaporate into steam. The water in the surface water layer gradually evaporates, and the impurities in the water are retained in the photothermal absorption layer because they cannot evaporate, thus achieving desalination. Multi-layer interfacial evaporators for efficient interfacial evaporation technology are attracting increased attention from researchers for the scientifically based optimization of their parameters such as nature, morphology, thickness, interface, and structure.”
Comments 4: Review objectives are vague and confusing and English is written poorly. Especially, on page 3, lines 79-83 don’t give clear meaning.
Response 4: Thanks very much for your comment. We are sorry for our careless mistakes about the language and writing style. We have received assistance from a professional group, the MDPI Editorial Team, and have carefully checked the manuscript to polish our revised version. All the errors you mentioned in your revision comments have been corrected, and we have also carefully checked and corrected the grammatical spelling of the remaining parts
The modified part is as follows:
Lines 81-83: “Compared with the core photothermal layer, the support layer has received less attention in many studies. However, with the deepening of research on the support layer, improving the support layer has gradually become an indispensable way to improve the overall performance of an SISG evaporator.”
The manuscript has also been carefully revised to minimized formatting issues and grammatical/typo errors. All the corrections were highlighted in Red for your easier reviewing in revised manuscript. In addition, the certification of English Editing, which got from the MDPI Editorial Team, has also been attached.
The modified errors are follows:
Line 24, “In addition” to “Additionally”
Line 25, “are” to “were”
Line 28, “to” to “for”
Line 30, “for the effectively respond” to “to enable effective responses”
Line 36, insert “and”
Line 37, delay“etc.”
Line 38, “the evaporation of water to generate steam” to “vapour”
Lines 38-39, “The condensed water collected afterward effectively desalinates seawater.” to “Collect the condensed water after evaporation to complete the desalination of sea water.”
Line 41, “In commonly” to “Generally”, “to” to “for”, “unitize” to “unitizing”
Line 42, “In commonly” to “Generally”
Lines 51-53, “Advanced fabrication of photothermal evaporator, including reasonable materials employment and optimized structure design, is the key to realize efficiency interfacial water evaporation.” to “The advanced fabrication of photothermal evaporators, including the employment of reasonable materials and optimized structure design, is the key to realizing efficient interfacial water evaporation.”
Line 54, “evaporator” to “evaporators”, “divided” to “classified”
Line 55, “evaporator” to “evaporators”
Line 56, insert “directly”
Line 60, insert “to”
Line 61, “Overall” to “overall”, “Material” to “material”, “Structural” to “structural”, “Scheme” to “scheme”, “Interfacial” to “interfacial”, “Solar” to “solar”, “Evaporation” to “evaporation”, “System” to “system”
Line 61, Figure 1, “integral heating” to “internal heating”
Line 81, “optimal” to “good”
Line 83, “of photothermal layer” to “on the support layer”, “the improving of the supporting layer” to “improving the support layer”
Line 85, insert “an”
Line 90, “insert “a”
Line 91, “investigates in one place” to “explores”
Line 94, insert “and”, delay “through current work”, “provide” to “describe”
Line 95, insert “the”
Line 100, “typically” to “typical”
Lines 102-103, delay “so the”, insert “Thus, the thermal conductivity and water absorption characteristics should be carefully considered when choosing the support layer material.”
Line 105, “polymer based” to “polymer-based”, “featured” to “characterized”
Line 107 “light buoyancy” delay “light”
Line 108, delay “Besides”, “aerogels” to “Aerogels”, insert “and they”
Line 109, “by” to “because of”
Line 111, “by” to “using”, “hold” to “possesses”, insert “a”
Line 112, “eco-friendly” to “eco-friendliness”
Line 114, “structure” to “structures”
Line 115, “on” to “of”, “modified by” to “and modification with”
Line 116, delay “and etc.”, “which make” to “enable”, insert “to”, “efficiency of evaporation” to “evaporation efficiency”
Line 117, “suitable” to “suitability”
Line 124, “well as” to “their”, “cost-effective” to “cost-effectiveness”
Line 125, “float” to “structure”
Line 108, “often exhibiting” to “which often exhibit”
Line 128, “after” to “loss”
Lines 129-130, “This insulation does the effective function to maintain high evaporation efficiency in the evaporator.” to “This insulation is responsible for effectively maintaining high evaporation efficiency in the evaporator.”
Lines 131-132, “In addition, the other important feature, porous structure, of commercial plastic foams further offers several advantages during application in SISG technology.” to “The porous structure of commercial plastic foams is another important feature that confers several advantages during application in SISG technology.”
Lines 135-136, “enabling more water absorption and further improving evaporation efficiency.” to “allowing more water to absorb and diffuse into capillary water flow further improving evaporation efficiency.”
Line 137, insert “it”
Line 140, “purposefully” to “effectively”
Line 142, “reached on” to “showed”, insert “a good”
Line 143-144, delay “and 83 %, respectively.”
Line 146, “Thi Kieu Trang Nguyen” to “Thi”
Line 150, “Besides” to “Additionally”, “Hiran D. Kiriarachchi” to “Hiran”
Line 152, “avoiding” to “preventing”
Line 153, “evaporating” to “evaporation”
Line 155, “Schematic” to “schematic”, “Thi Kieu Trang Nguyen” to “Thi”
Line 156, “Schematic” to “schematic”, “Hiran D. Kiriarachchi” to “Hiran”
Line 157, delay “However”, “due” to “Due”, “accelerated by a lot of” to “achieved in many recent”
Line 158, “been already greater than” to “long surpassed”, “speed” to “rate”
Line 159, insert “the”
Line 160, “have been done” to “has been performed”
Line 165, “carried out” to “involved”
Line 166, insert “of”
Line 168, “of” to “in”
Line 170, “took away some heat from” to “resulted in some heat leaving”, “it” to “This”
Lines 171-172, delay “and the evaporation of evaporator water increases linearly.”
Line 174, “by” to “in terms of”, “polluted environments” to “pollution affect”
Line 177, “corrosion resistance” to “corrosion-resistant”
Line 178, “aim” to “have aimed”
Line 180, “Besides” to “Additionally”, “pre to foam” to “prior to the foaming”
Line 183, “high performance” to “high-performance”, “are” to “is”
Line 185, “have better” to “improve”
Line 187, “allows the foam to have better” to “usually improves the”, insert “of the foam”
Line 190, “also selected as an well” to “is also considered a good”
Line 192, insert “and is”
Line 199, “is” to “was”
Line 200, “the” to “a”
Line 201, “adhere” to “adhered”
Line 204, “reaching below the safe” to “and the level of mercury fell below the safety”
Line 205, “mercury” to “water”
Line 206, insert “an”
Line 207, “absorption for” to “capacity to absorb”
Line 211, “requires” to “necessitates”
Line 217, “the” to “an”, “of: to “for”, “is” to “of”
Line 218, insert “was achieved”, “is up to” to “reached”
Line 226, “By” to “In”
Line 228, “nano Ag-based” to “nano-Ag-based”
Line 230, insert “and”
Line 231, “is” to “was”
Line 232, insert “and”
Line 233, “has” to “had”
Line 236, “have” to “has”, “many researches” to “much research”
Line 238, “But” to “However”, insert “a”
Line 239, “facing” to “presents”, “process” to “processes”
Line 240, “as well as difficult to degrade which is unfriendly to environment” to “and difficulty in degradation, making it unfriendly to the environment”
Line 243, “Gel” to “a gel”, insert “a”
Line 245, “supporting layer” to “support layers”
Line 246, “gel” to “gels”
Line 247, “by” to “with”
Line 251, “it” to “they”, insert “quickly”, delay “timely”
Line 253, “a lot of” to “significant”
Line 254, “when” to “during”, “based on” to “using”
Line 258, insert “a”, insert “good”
Line 259, “agent” to “agents”
Line 262, “to enhance” to “that enhanced”
Line 265, “reaches up to” to “reached”, “is” to “was”, insert “the”
Line 276, “research” to “researchers”, “hydrogels” to “hydrogels’ properties”
Line 277, insert “their”, “bacteriostasis” to “and bacteriostatic effects”
Line 278, “which make hydrogels have a wider use prospect.” to “to provide hydrogels with more prospective uses.”
Line 280, “The” to “A”
Line 281, “by” to “through”, “The” to “A”
Line 282, “network” to “network’s”
Line 286, insert “the”
Line 288, “low density” to “low-density”
Line 289, “high surface area” to “high-surface-area”
Line 290, “researches” to “research”
Line 293, insert “and”
Line 304, insert “an”
Line 306, “What's more” to “Furthermore”
Line 308, insert “them”
Line 310, “straw derived” to “straw-derived”
Line 317, “itself does not” to “is not”, “possess the ability of” to “able to perform”
Line 318, “has high scalability” to “is highly scalable”
Lines 322-323, “it should be considered to improve its durability and mechanical properties during preparation.” to “steps to improve its durability and mechanical properties during preparation should be considered.”
Line 326, “Schematic” to “schematic”
Line 330, “with” to “have the”
Line 331, insert “and”
Line 332, insert “a”
Line 335, insert “the”
Line 336, “time” to “periods of”
Line 340, insert “has been conducted”, “as support layer directly” to “has been directly used as a support layer”
Line 343, “exhibit” to “exhibits”
Line 347, insert “a”, “by” to “through”
Line 351, insert “the”, “in high salinity” to “to high-salinity”
Line 354, “of” to “in the”, “wood based” to “wood-based”
Line 358, “by in-situ” to “through the”
Line 359, “is” to “was”
Line 360, “high-light” to “very light”
Line 377, insert “the”, “on nature plant based” to “in natural plant-based”
Line 379, insert “and”, delay “etc.”, “used in the study” to “studied”
Line 381, insert “The”, “make the material have enhanced” to “enhance the material’s”
Lines 383-384, “so as to avoid re-designing” to “avoiding the need to re-design”
Lines 384-385, “by this operation have the ability of photothermal conversion” to “in this manner has a photothermal conversion ability”
Line 386, “like” to “such as”
Line 390, “as well as” to “and”
Line 394, “light absorbing” to “light-absorbing”
Line 400, “Subsequent” to “The subsequent carbonization”
Line 401, “continue to improve the” to “further improve their”, insert “used”
Line 405, “provides” to “enables”
Line 408, insert “and”, delay “so on”
Line 417, “by” to “through”
Line 418, “small” to “low”
Line 419, “make” to “provide”, “multi-functions” to “multiple functions”
Line 420, “edge” to “frontier”
Line 425, “Due to” to “Under”
Line 426, “will be” to “can be”, delay “to a certain extent”, “Thereby, water transport capacity inside the material was affected.” to “affecting the water transport capacity inside the material.”
Line 428, “capacity” to “capacities”
Line 433, insert “and”
Line 434, “easy to degrade without pollution to the environment” to “degrades easily without polluting the environment”, insert “an”
Line 435, “by in-situ” to “using the in situ”
Line 436, insert “a”
Line 448, “exhibit” to “exhibits”
Line 451, “is” to “was”
Line 454, “experiment” to “experimental”
Line 458, “as well as” to “and”
Line 459, “in above ways” to “in the ways mentioned above”
Line 480, insert “which”, “plays a crucial role in” to “is crucial for”
Line 481, insert “the”
Line 484, “are” to “have been”, “for example” to “such as”
Line 485, “structure” to “structures”, insert “and”
Line 486, insert “mainly based on polymers”, insert “being”
Line 491, “possibility” to “probability of”, “will be” to “being”
Line 493, insert “the”
Line 500, “air planking” to “air-planking”
Line 505, “By” to “When”
Line 506, “are up to” to “reached”
Line 507, “cotton based” to “cotton-based”
Line 508, “Cone” to “cone”, “is” to “was”
Line 548, “superhydrophile” to “super hydrophile”
Line 551, “3D” to “Three-dimensional”
Line 552, “process of producing” to “production of”
Line 558, “of” to “in the”
Line 566 “including” to “include”
Line 571, “yet their scalability” to “yet scaling them”
Line 589, “improve” to “improving”, “make” to “making”
Comments 5: Usually Abstract represents what you have done/written in the article and the conclusion depicts your findings/what you have achieved after doing so much literature on said topic. In this perspective, I would suggest authors to rewrite the abstract and conclusions sections.
Response 5: Thank you for pointing this out. We agree with this comment. We have rewritten the abstract and conclusion of the review.
The abstract is supplemented as follows:
“With the increasing shortage of water resources and the aggravation of water pollution, solar-driven interfacial steam generation (SISG) technology has garnered considerable attention because of its low energy consumption, simple operation, and environmental friendliness. The popular multi-layer SISG evaporator is composed of two basic structures: a photothermal layer and a support layer. Herein, the support layer underlies the photothermal layer and carries out thermal management, supports the photothermal layer, and transports water to the evaporation interface to improve the stability of the evaporator. While most research is focuses on the photothermal layer, the support layer is typically viewed as a supporting object for the photothermal layer. This review focuses on the support layer, which is relatively neglected in evaporator development. It summarizes existing researches in the field of multi-layer interface evaporators, based on various of polymers and biomaterials, as along with their advantages and disadvantages. Specifically, polymer-based support layers are mainly reviewed, including polymer foams, gels, and their corresponding functional materials. While biomaterial support layers, including natural plants, carbonized biomaterials, and other innovation biomaterials. Additionally, the corresponding structure design strategies for the support layer were also involved. It was found that the selection and optimal design of the substrate also play an important role in the efficient operation of the whole steam generation system. Their evolution and refinement are vital for advancing the sustainability and effectiveness of interfacial evaporation technology. The corresponding potential future research direction and application prospects of support layer materials are carefully presented to enable effective responses to global water challenges.”
The conclusion is supplemented as follows:
“In this paper, material selection and structural design of support layer in the field of multi-layer SISG progress are reviewed. Support layer material selection focuses on polymers and biomaterials. The polymers, which are mainly chosen as support layer materials, include foams and gels, while the biomaterials include natural plants, carbonized, and further modified biomaterials. Original unmodified materials like polymer foams and natural plants offer simplicity in preparation, they often lack durability for practical applications, especially in harsh environments such as high-salinity conditions. Modified support layer materials have shown promise in enhancing performance to some extent, yet scaling them from laboratory settings to practical large-scale applications remains a challenge. In addition to the innovation of supporting layer material selection, recent research progress emphasizes the fact that structural optimization is also a key way to improve the efficiency of photothermal evaporators. Innovative 3D support layer structures, such as cone and arch designs, effectively regulate the light absorption area and steam flow resistance, presenting new avenues for surface structure engineering in enhancing photothermal evaporator performance.
Summarizing the existing research progress, we propose that the most effective support layer structure should maintain a balance between the choice of materials and their potential for wide application. However, the design of the support layer to achieve this balance still faces many challenges. Compared with photothermal materials, the selection of support layer materials is relatively limited, and most studies focus on solid materials. With the deepening of research, simple seawater desalination can no longer meet the current demand for clean energy. The field of water treatment in the real world is much more complex. The support layer is not limited to the role of water transport support. The functionalization and integration of the support layer have become the focus of the subsequent design of photothermal evaporators. Broadening the selection of materials, improving the performance of the support layer, and making it commercially applicable will need to be considered in future research. It is believed that SISG technology, supported by various advanced support layers, can help drive eco-friendly, cost-effective, and sustainable development in the field of water purification.”
Comments 6: Please arrange all the references into a single format. e.g. journal name in full form or abbreviated form. E.g., References 30 and 36 are from the same journal but the names are in a different format. In reference 39 abbreviation is used for the journal name, please check all the references in this context.
Response 6: Thank you for pointing this out. We have corrected the errors you mentioned and have carefully checked the format of all references so that they meet the required format for MDPI journals.
All the modified references are listed as below:
[30] Wang, L. Zhao, F. Zhang, K. Yu, C. Yang, J. Jia, W. Guo, J. Zhao, F. Qu, Synthesis of a Co-Sn Alloy-Deposited PTFE Film for Enhanced Solar-Driven Water Evaporation via a Super-Absorbent Polymer-Based "Water Pump" Design, ACS Appl. Mater. Interfaces, 13 (2021) 26879-26890.
[36] J. Pinto, D. Magrì, P. Valentini, F. Palazon, J.A. Heredia-Guerrero, S. Lauciello, S. Barroso-Solares, L. Ceseracciu, P.P. Pompa, A. Athanassiou, D. Fragouli, Antibacterial Melamine Foams Decorated with in Situ Synthesized Silver Nanoparticles, ACS Appl. Mater. Interfaces, 10 (2018) 16095-16104.
Thanks again for your carefully review and valuable comments, which improved the quantity of our manuscript highly! Sincerely thanks!

Reviewer 3 Report
Comments and Suggestions for Authors
The problem of fresh water shortage, which is growing from year to year, requires the development of new technologies, including seawater desalination. The technologies used for this are often very material and energy consuming. One of the promising technologies in this regard is the use of solar energy as a renewable heat source for the evaporation and subsequent condensation of water. Among the strategies used the most promising is interfacial heating having an advantage to focus the heat at the surface-air interface. Multicomponent systems on the water surface used for this purpose, including a thermal layer and a substrate as the main components, attract increased attention from researchers for scientifically based optimization of their parameters - nature, morphology, thickness, interface, structure. In this case, the substrate is often considered only as a supporting object for the thermal layer. The present review, which summarizes the studies in this area, comes to the clear conclusion that the choice of substrate plays a significant role in the efficient operation of the entire steam generation system. It is shown that it is necessary to maintain a balance between the selection of the most effective structure and morphology of the substrate and the possibility of its wide practical application. It is shown that by selecting the three-dimensional structure of the substrate, the light absorption area and steam flow resistance can be adjusted in a controlled manner.
The strength of the work: Quite complete critical overview of the studies in the field, highlighting the importance of the subject of the support of SISG systems, previously been underestimated. The prospects for the most effective development of this direction for practical eco-friendly application through the selection of substrate parameters are outlined.
The weakness: The work would gain if the most relevant one or two case studies, stressing the conclusions of the present work, were overviewed in more detail.
In general, the review is scientifically sound with the appropriate design to address the issues under consideration. It is clear to follow, relevant for the field and presented in a well-structured manner. The manuscript provides sufficient details that support the conclusions. Referencing is quite comprehensive and up-to-dated. The manuscript is suitable for publication in Polymers in its present form.
The technical points to correct:
1) Images 1 and 2 in Fig. 1 should be swapped to match the text (lines 40-43), or vice versa.
2) The text should be checked for misprints, punctuation and small corrections:
Line 57: “this simplicity often leads poor durability” - “… leads to poor …”
Line 89: “typically design” – “typical design”.
etc.
Author Response
General comment: The weakness: The work would gain if the most relevant one or two case studies, stressing the conclusions of the present work, were overviewed in more detail.
Response: Thank you for pointing this out. In this review, we mentioned “The advanced fabrication of photothermal evaporators, including the employment of reasonable materials and optimized structure design, is the key to realizing efficient interfacial water evaporation.”However, the literature on the optimization of the structure of the photothermal evaporator quoted in this paper is not detailed enough. Therefore, we have made a supplement to the description of the optimal design of the structure in the list.
The modified part is as follows:
Lines 535-543, insert “Wu et al. [91] developed a pyramid evaporator using 3D printing technology (Figure 8d). Surface distributed micropores are formed on the prepared surface, endowing the pyramid evaporator with ultra-fast water spreading property. Due to the designed morphology of the 3D structure with asymmetric grooves and the gradient microcavity arrays, the liquid film spreads on the structure surface displays position-related liquid film thickness and temperature gradient along the sidewall, which further leads to the thermocapillary force inside the liquid film and the capability to capture energy from the surrounding environment to enhance water evaporation and energy efficiency.”
At the same time, we have briefly made a summary table of different types of support layers materials in multi-layered interfacial evaporators, including the typical materials and main advantages. The table was named Table 1 and present it on page 3.
Table 1. The brief review of different types of support layer material in multi-layer SISG evaporators.
Support layer category |
Typical materials |
Main advantages |
Commercial Polymer Foam |
Polyurethane (PU) foams Polystyrene (PS) foams Polydimethylsiloxane (PDMS) foam |
Light weight, self-floating, low thermal, conductivity. |
Functional Polymer Foam |
Coating Functionalized Polymer Foam Substrate Modified Functional Polymer Foam |
Light weight, self-floating, high-performance, multi-functionalization. |
Gels |
Hydrogel Aerogel |
Environmentally friendly, unique transporting water structure. |
Biomaterials |
Natural Plants Carbonized Biomaterials Modified Biomaterials |
Cost-effective, rich source environmentally friendly. |
Comments 1: Images 1 and 2 in Fig. 1 should be swapped to match the text (lines 40-43), or vice versa.
Response 1: Thanks very much for your comment. We have swapped the order of presentation on photothermal evaporators in the text to match image1 and 2 in Figure 1. (Line 43-47)
The modified part is as follows:
“1) evaporation based on internal heating, in which uniformly dispersed solar absorbers convert incident solar photons into thermal energy to heat the liquid; 2) evaporation based on bottom heating, in which solar energy is absorbed by a solar absorber and converted into thermal energy to heat a large amount of liquid at the bottom of the evaporator;”
Comments 2: The text should be checked for misprints, punctuation and small corrections
Line 57: “this simplicity often leads poor durability” - “… leads to poor …”
Line 89: “typically design” – “typical design”.
etc.
Response 2: Thanks very much for your comment. We are sorry for our careless mistakes about the language and writing style. We have received assistance from a professional group, the MDPI Editorial Team, and have carefully checked the manuscript to polish our revised version. All the errors you mentioned in your revision comments have been corrected, and we have also carefully checked and corrected the grammatical spelling of the remaining parts
The manuscript has been carefully revised to minimized formatting issues and grammatical/typo errors. All the corrections were highlighted in Red for your easier reviewing in revised manuscript. In addition, the certification of English Editing, which got from the MDPI Editorial Team, has also been attached.
The modified errors are as follows:
Line 24, “In addition” to “Additionally”
Line 25, “are” to “were”
Line 28, “to” to “for”
Line 30, “for the effectively respond” to “to enable effective responses”
Line 36, insert “and”
Line 37, delay“etc.”
Line 38, “the evaporation of water to generate steam” to “vapour”
Lines 38-39, “The condensed water collected afterward effectively desalinates seawater.” to “Collect the condensed water after evaporation to complete the desalination of sea water.”
Line 41, “In commonly” to “Generally”, “to” to “for”, “unitize” to “unitizing”
Line 42, “In commonly” to “Generally”
Lines 51-53, “Advanced fabrication of photothermal evaporator, including reasonable materials employment and optimized structure design, is the key to realize efficiency interfacial water evaporation.” to “The advanced fabrication of photothermal evaporators, including the employment of reasonable materials and optimized structure design, is the key to realizing efficient interfacial water evaporation.”
Line 54, “evaporator” to “evaporators”, “divided” to “classified”
Line 55, “evaporator” to “evaporators”
Line 56, insert “directly”
Line 60, insert “to”
Line 61, “Overall” to “overall”, “Material” to “material”, “Structural” to “structural”, “Scheme” to “scheme”, “Interfacial” to “interfacial”, “Solar” to “solar”, “Evaporation” to “evaporation”, “System” to “system”
Line 61, Figure 1, “integral heating” to “internal heating”
Line 81, “optimal” to “good”
Line 83, “of photothermal layer” to “on the support layer”, “the improving of the supporting layer” to “improving the support layer”
Line 85, insert “an”
Line 90, “insert “a”
Line 91, “investigates in one place” to “explores”
Line 94, insert “and”, delay “through current work”, “provide” to “describe”
Line 95, insert “the”
Line 100, “typically” to “typical”
Lines 102-103, delay “so the”, insert “Thus, the thermal conductivity and water absorption characteristics should be carefully considered when choosing the support layer material.”
Line 105, “polymer based” to “polymer-based”, “featured” to “characterized”
Line 107 “light buoyancy” delay “light”
Line 108, delay “Besides”, “aerogels” to “Aerogels”, insert “and they”
Line 109, “by” to “because of”
Line 111, “by” to “using”, “hold” to “possesses”, insert “a”
Line 112, “eco-friendly” to “eco-friendliness”
Line 114, “structure” to “structures”
Line 115, “on” to “of”, “modified by” to “and modification with”
Line 116, delay “and etc.”, “which make” to “enable”, insert “to”, “efficiency of evaporation” to “evaporation efficiency”
Line 117, “suitable” to “suitability”
Line 124, “well as” to “their”, “cost-effective” to “cost-effectiveness”
Line 125, “float” to “structure”
Line 108, “often exhibiting” to “which often exhibit”
Line 128, “after” to “loss”
Lines 129-130, “This insulation does the effective function to maintain high evaporation efficiency in the evaporator.” to “This insulation is responsible for effectively maintaining high evaporation efficiency in the evaporator.”
Lines 131-132, “In addition, the other important feature, porous structure, of commercial plastic foams further offers several advantages during application in SISG technology.” to “The porous structure of commercial plastic foams is another important feature that confers several advantages during application in SISG technology.”
Lines 135-136, “enabling more water absorption and further improving evaporation efficiency.” to “allowing more water to absorb and diffuse into capillary water flow further improving evaporation efficiency.”
Line 137, insert “it”
Line 140, “purposefully” to “effectively”
Line 142, “reached on” to “showed”, insert “a good”
Line 143-144, delay “and 83 %, respectively.”
Line 146, “Thi Kieu Trang Nguyen” to “Thi”
Line 150, “Besides” to “Additionally”, “Hiran D. Kiriarachchi” to “Hiran”
Line 152, “avoiding” to “preventing”
Line 153, “evaporating” to “evaporation”
Line 155, “Schematic” to “schematic”, “Thi Kieu Trang Nguyen” to “Thi”
Line 156, “Schematic” to “schematic”, “Hiran D. Kiriarachchi” to “Hiran”
Line 157, delay “However”, “due” to “Due”, “accelerated by a lot of” to “achieved in many recent”
Line 158, “been already greater than” to “long surpassed”, “speed” to “rate”
Line 159, insert “the”
Line 160, “have been done” to “has been performed”
Line 165, “carried out” to “involved”
Line 166, insert “of”
Line 168, “of” to “in”
Line 170, “took away some heat from” to “resulted in some heat leaving”, “it” to “This”
Lines 171-172, delay “and the evaporation of evaporator water increases linearly.”
Line 174, “by” to “in terms of”, “polluted environments” to “pollution affect”
Line 177, “corrosion resistance” to “corrosion-resistant”
Line 178, “aim” to “have aimed”
Line 180, “Besides” to “Additionally”, “pre to foam” to “prior to the foaming”
Line 183, “high performance” to “high-performance”, “are” to “is”
Line 185, “have better” to “improve”
Line 187, “allows the foam to have better” to “usually improves the”, insert “of the foam”
Line 190, “also selected as an well” to “is also considered a good”
Line 192, insert “and is”
Line 199, “is” to “was”
Line 200, “the” to “a”
Line 201, “adhere” to “adhered”
Line 204, “reaching below the safe” to “and the level of mercury fell below the safety”
Line 205, “mercury” to “water”
Line 206, insert “an”
Line 207, “absorption for” to “capacity to absorb”
Line 211, “requires” to “necessitates”
Line 217, “the” to “an”, “of: to “for”, “is” to “of”
Line 218, insert “was achieved”, “is up to” to “reached”
Line 226, “By” to “In”
Line 228, “nano Ag-based” to “nano-Ag-based”
Line 230, insert “and”
Line 231, “is” to “was”
Line 232, insert “and”
Line 233, “has” to “had”
Line 236, “have” to “has”, “many researches” to “much research”
Line 238, “But” to “However”, insert “a”
Line 239, “facing” to “presents”, “process” to “processes”
Line 240, “as well as difficult to degrade which is unfriendly to environment” to “and difficulty in degradation, making it unfriendly to the environment”
Line 243, “Gel” to “a gel”, insert “a”
Line 245, “supporting layer” to “support layers”
Line 246, “gel” to “gels”
Line 247, “by” to “with”
Line 251, “it” to “they”, insert “quickly”, delay “timely”
Line 253, “a lot of” to “significant”
Line 254, “when” to “during”, “based on” to “using”
Line 258, insert “a”, insert “good”
Line 259, “agent” to “agents”
Line 262, “to enhance” to “that enhanced”
Line 265, “reaches up to” to “reached”, “is” to “was”, insert “the”
Line 276, “research” to “researchers”, “hydrogels” to “hydrogels’ properties”
Line 277, insert “their”, “bacteriostasis” to “and bacteriostatic effects”
Line 278, “which make hydrogels have a wider use prospect.” to “to provide hydrogels with more prospective uses.”
Line 280, “The” to “A”
Line 281, “by” to “through”, “The” to “A”
Line 282, “network” to “network’s”
Line 286, insert “the”
Line 288, “low density” to “low-density”
Line 289, “high surface area” to “high-surface-area”
Line 290, “researches” to “research”
Line 293, insert “and”
Line 304, insert “an”
Line 306, “What's more” to “Furthermore”
Line 308, insert “them”
Line 310, “straw derived” to “straw-derived”
Line 317, “itself does not” to “is not”, “possess the ability of” to “able to perform”
Line 318, “has high scalability” to “is highly scalable”
Lines 322-323, “it should be considered to improve its durability and mechanical properties during preparation.” to “steps to improve its durability and mechanical properties during preparation should be considered.”
Line 326, “Schematic” to “schematic”
Line 330, “with” to “have the”
Line 331, insert “and”
Line 332, insert “a”
Line 335, insert “the”
Line 336, “time” to “periods of”
Line 340, insert “has been conducted”, “as support layer directly” to “has been directly used as a support layer”
Line 343, “exhibit” to “exhibits”
Line 347, insert “a”, “by” to “through”
Line 351, insert “the”, “in high salinity” to “to high-salinity”
Line 354, “of” to “in the”, “wood based” to “wood-based”
Line 358, “by in-situ” to “through the”
Line 359, “is” to “was”
Line 360, “high-light” to “very light”
Line 377, insert “the”, “on nature plant based” to “in natural plant-based”
Line 379, insert “and”, delay “etc.”, “used in the study” to “studied”
Line 381, insert “The”, “make the material have enhanced” to “enhance the material’s”
Lines 383-384, “so as to avoid re-designing” to “avoiding the need to re-design”
Lines 384-385, “by this operation have the ability of photothermal conversion” to “in this manner has a photothermal conversion ability”
Line 386, “like” to “such as”
Line 390, “as well as” to “and”
Line 394, “light absorbing” to “light-absorbing”
Line 400, “Subsequent” to “The subsequent carbonization”
Line 401, “continue to improve the” to “further improve their”, insert “used”
Line 405, “provides” to “enables”
Line 408, insert “and”, delay “so on”
Line 417, “by” to “through”
Line 418, “small” to “low”
Line 419, “make” to “provide”, “multi-functions” to “multiple functions”
Line 420, “edge” to “frontier”
Line 425, “Due to” to “Under”
Line 426, “will be” to “can be”, delay “to a certain extent”, “Thereby, water transport capacity inside the material was affected.” to “affecting the water transport capacity inside the material.”
Line 428, “capacity” to “capacities”
Line 433, insert “and”
Line 434, “easy to degrade without pollution to the environment” to “degrades easily without polluting the environment”, insert “an”
Line 435, “by in-situ” to “using the in situ”
Line 436, insert “a”
Line 448, “exhibit” to “exhibits”
Line 451, “is” to “was”
Line 454, “experiment” to “experimental”
Line 458, “as well as” to “and”
Line 459, “in above ways” to “in the ways mentioned above”
Line 480, insert “which”, “plays a crucial role in” to “is crucial for”
Line 481, insert “the”
Line 484, “are” to “have been”, “for example” to “such as”
Line 485, “structure” to “structures”, insert “and”
Line 486, insert “mainly based on polymers”, insert “being”
Line 491, “possibility” to “probability of”, “will be” to “being”
Line 493, insert “the”
Line 500, “air planking” to “air-planking”
Line 505, “By” to “When”
Line 506, “are up to” to “reached”
Line 507, “cotton based” to “cotton-based”
Line 508, “Cone” to “cone”, “is” to “was”
Line 548, “superhydrophile” to “super hydrophile”
Line 551, “3D” to “Three-dimensional”
Line 552, “process of producing” to “production of”
Line 558, “of” to “in the”
Line 566 “including” to “include”
Line 571, “yet their scalability” to “yet scaling them”
Line 589, “improve” to “improving”, “make” to “making”
Thanks again for your carefully review and valuable comments, which improved the quantity of our manuscript highly! Sincerely thanks!

Round 2
Reviewer 1 Report
Development Status of SISG Support Layer Based on Polymers and Biomaterials
The reviewed work discusses the use of various materials to create a supporting layer in solar-driven interfacial steam generation (SISG). Although the word Polymer appears in the title, the work does not consider methods of creating polymers intended for this application or their specific properties. Looking at the entire manuscript, the appropriate place for publication will be MDPI Energies.
A number of language errors have been removed in the new version of the manuscript in accordance with the reviewers' suggestions. The descriptions of SISG as a system have also been expanded.
Due to the lack of own experimental studies, the work is rather a typical review paper and this should be noted in the title. The title should also expand the abbreviation SISG, which is not in common use. In the context of polymer science, the acronym SISG stands for Superimposed Isoprene-Styrene Gradient. It refers to a specific type of gradient block copolymer that consists of isoprene and styrene segments.
The work should include a more detailed discussion of the principles of operation of the entire SIGS system, not just the support layer.
In many places, authors provide the first, middle and third names and surname of the first author (Thi Kieu Trang Nguyen et al.), while the common practice is to provide only the surname. By the way, et al. It should be written in italics because it comes from Latin.
Examples:
Line 192 - Javier Pinto et al. [36]
Line 302 - Daniel Peter Storer et al. [54]
“The supporting layer is made of cylindrical hollow foam with a diameter of 32 mm. SAP particles are added into the hollow foam.”? What is the structure of the mentioned hollow foam? Is it something like a 32mm diameter cylindrical sponge or a collection of parallel tubes? If these are pipes, what diameter are they?
There is a passage in Conclusion and Prospect that makes no specific sense: "Summarizing the existing research progress, we propose that the most effective support layer structure should maintain a balance between the choice of materials and their potential for wide application."
Similarly, the next sentence makes no sense in the context of this manuscript: “With the deepening of research, simple seawater desalination can no longer meet the current demand for clean energy.”
Comments on the Quality of English Languageet al. It should be written in italics because it comes from Latin.
Reformat list of References; References
1. [1] P. Tao, G. Ni, C. Song, W. Shang, J. Wu, J. Zhu, G. Chen, T. Deng, Solar-driven interfacial evaporation, Nature Energy, 3 (2018) 1031-1041.
2. [2] M. Kurihara, Seawater Reverse Osmosis Desalination, Membranes, 11 (2021).
3. [3] E. G. de Moraes, L. Sangiacomo, N. P. Stochero, S. Arcaro, L. R. Barbosa, A. Lenzi, C. Siligardi, A.P. Novaes de Oliveira, Innovative thermal and acoustic insulation foam by using recycled ceramic shell and expandable styrofoam (EPS) wastes, Waste Manage. Res., 89 (2019) 336-344.
etc. etc.
Author Response
General comment: The reviewed work discusses the use of various materials to create a supporting layer in solar-driven interfacial steam generation (SISG). Although the word Polymer appears in the title, the work does not consider methods of creating polymers intended for this application or their specific properties. Looking at the entire manuscript, the appropriate place for publication - after removing the errors, will be MDPI Energies.
Response: Thanks for your suggestion, we have already made the modification in the first round of revisions and attached in the response to your comment. The important of polymers in the revised manuscript by modifying corresponding sections, especially in section of Abstract, Introduction and Conclusion. Thanks again for your suggestion which highly improve our manuscript to meet the scope of MDPI Polymers.
Comments 1: Due to the lack of own experimental studies, the work is rather a typical review paper and this should be noted in the title. The title should also expand the abbreviation SISG, which is not in common use. In the context of polymer science, the acronym SISG stands for Superimposed Isoprene-Styrene Gradient. It refers to a specific type of gradient block copolymer that consists of isoprene and styrene segments.
Response 1: Thanks for your comment, we have revised the title of the article according to your comments. The modification is also highlighted in Blue in revised manuscript for your checking.
The revised title is as follows:
Development Status of Solar-driven Interfacial Steam Generation Support Layer Based on Polymers and Biomaterials: A Review
On the question of SISG abbreviation, SISG is the abbreviation of solar-driven interfacial steam generation, which is recognized and widely used in articles about interfacial evaporation. We are sorry for making the confusion.
Comments 2: In many places, authors provide the first, middle and third names and surname of the first author (Thi Kieu Trang Nguyen et al.), while the common practice is to provide only the surname. By the way, et al. It should be written in italics because it comes from Latin.
Response 2: Thank you very much for your comments, we have carefully revised the referred articles according to your comments. The modifications are also highlighted in Blue in revised manuscript for your checking.
The modified part is as follows:
Line 137, “Shi et al.” to “Shi et al.”
Line 147, “Thi et al.” to “Nguyen et al.”
Line 152, “Hiran et al.” to “Kiriarachchi et al.”
Line 156, “Thi” to “Nguyen”
Line 157, “Hiran” to “Kiriarachchi”
Line 198, “Javier Pinto” to “Pinto et al.”
Line 214, “Shi et al.” to “Shi et al.”
Line 230, “Wang et al.” to “Wang et al.”
Line 259, “Shi et al.” to “Shi et al.”
Line 267, “Wang et al.” to “Wang et al.”
Line 299, “Wu et al.” to “Wu et al.”
Line 311, “Wang et al.” to “Storer et al.”
Line 343, “Chen et al.” to “Chen et al.”
Line 357, “Shi et al.” to “Shi et al.”
Line 365, “Zhang et al.” to “Zhang et al.”
Line 373, “Fang et al.” to “Fang et al.”
Line 393, “Chen et al.” to “Chen et al.”
Line 399, “Xu et al.” to “Xu et al.”
Line 403, “Zeng et al.” to “Zeng et al.”
Line 432, “Guo et al.” to “Guo et al.”
Line 448, “Zhang et al.” to “Zhang et al.”
Comments 3: “The supporting layer is made of cylindrical hollow foam with a diameter of 32 mm. SAP particles are added into the hollow foam.”? What is the structure of the mentioned hollow foam? Is it something like a 32mm diameter cylindrical sponge or a collection of parallel tubes? If these are pipes, what diameter are they?
Response 3: Thank you very much for your question. In order to answer your question, we have consulted the contents of the original literature, and the answers are as follows:
The supporting layer here consists of a cylindrical hollow foam and a SPA hyper absorbent material that fills the hollow part of the foam. The diameter of the cylindrical foam is 32 mm, and the diameter of the hollow part of the foam is 12 mm. The modification is also highlighted in Blue in revised manuscript for your checking.
The original description also been modified as follows:
“A cylindrical foam with a diameter of 32 mm was utilized for the floating substrate. The inside of the foam is hollow, and a 12 mm diameter hole was cut out at the bottom of the foam.”
Comments 4: There is a passage in Conclusion and Prospect that makes no specific sense: "Summarizing the existing research progress, we propose that the most effective support layer structure should maintain a balance between the choice of materials and their potential for wide application."
Similarly, the next sentence makes no sense in the context of this manuscript: “With the deepening of research, simple seawater desalination can no longer meet the current demand for clean energy.”
Response 4: Thank you very much for your comments, we have canceled your mentioned sentences and revised the Conclusion of the article according to your comments. The modification is also highlighted in Blue in revised manuscript for your checking.
The modified part is as follows:
Lines 581-583: “The design of the support layer of the photothermal evaporator still faces many challenges. Compared with photothermal materials, the selection of support layer materials is relatively limited, and most studies focus on solid materials. The field of water treatment in the real world is much more complex.”
Comments 5: Reformat list of References
Response 5: Thanks for your comments, we have revised the reference format for this article in accordance with the reference format required by the MDPI journal. All the revised references are also highlighted in Blue.
Modified as follows:
- Tao, P.; Ni, G.; Song, C.; Shang, W.; Wu, J.; Zhu, J.; Chen, G.; Deng, T. Solar-driven interfacial evaporation. Nat.Energy 2018, 3, 1031-1041. https://doi.org/10.1038/s41560-018-0260-7
- Kurihara, M. Seawater Reverse Osmosis Desalination. Membranes2021, 11. https://doi.org/10.3390/membranes11040243
- de Moraes, E.; Sangiacomo, L.; P. Stochero, N.; Arcaro, S.; R. Barbosa, L.; Lenzi, A.; Siligardi, C.; Novaes de Oliveira, A.P. Innovative thermal and acoustic insulation foam by using recycled ceramic shell and expandable styrofoam (EPS) wastes. Waste Manage.2019, 89, 336-344. https://doi.org/10.1016/j.wasman.2019.04.019
- Lee, T.; Rahardianto, A.; Cohen, Y. Flexible reverse osmosis (FLERO) desalination. Desalination2019, 452, 123-131. https://doi.org/10.1016/j.desal.2018.10.022
- Abid, M.B.; Wahab, R.A.; Salam, M.A.; Moujdin, I.A.; Gzara, L. Desalination technologies, membrane distillation, and electrospinning, an overview. Heliyon2023, 9, e12810. https://doi.org/10.1016/j.heliyon.2023.e12810
- Alawad, S.M.; Khalifa, A.E. Analysis of water gap membrane distillation process for water desalination. Desalination2019, 470, 114088. https://doi.org/10.1016/j.desal.2019.114088
- Liu, X.; Li, L.; Wang, M.; Wang, D.; Yan, H.; Li, K.; Li, Y.; Yang, Y.; You, Y.; Yang, X.; et al. In-situ polymerization of PANI nanocone array on PEN nanofibrous membranes for solar-driven interfacial evaporation. Purif. Technol.2024, 344, 127109. https://doi.org/10.1016/j.seppur.2024.127109
- Abdelkader, B.A.; Antar, M.A.; Khan, Z. Nanofiltration as a Pretreatment Step in Seawater Desalination: A Review. Eng. 2018, 43, 4413-4432. https://doi.org/10.1007/s13369-018-3096-3
- Solouki, S.; Karrabi, M.; Eftekhari, M. Application of a functionalized thin-film composite nanofiltration membrane in water desalination. Mol. Liq. 2024, 399, 124399. https://doi.org/10.1016/j.molliq.2024.124399
- Sharshir, S.W.; Algazzar, A.M.; Elmaadawy, K.A.; Kandeal, A.W.; Elkadeem, M.R.; Arunkumar, T.; Zang, J.; Yang, N. New hydrogel materials for improving solar water evaporation, desalination and wastewater treatment: A review. Desalination2020, 491, 114564. https://doi.org/10.1016/j.desal.2020.114564
- Wang, J.; Kong, Y.; Liu, Z.; Wang, H. Solar-driven interfacial evaporation: Design and application progress of structural evaporators and functional distillers. Nano Energy2023, 108, 108115. https://doi.org/10.1016/j.nanoen.2022.108115
- Zhao, Q.; Yang, Y.; Pan, C.; Zhu, B.; Sha, Z.; Wei, Y.; Que, W. Integrated strategy of solar evaporator and steam collector configurations for interfacial evaporation water purification. Energy2023, 266, 112187. https://doi.org/10.1016/j.solener.2023.112187
- Lu, X.; Mu, C.; Liu, Y.; Wu, L.; Tong, Z.; Huang, K. Recent advances in solar-driven interfacial evaporation coupling systems: Energy conversion, water purification, and seawater resource extraction.Nano Energy 2024, 120, 109180. https://doi.org/10.1016/j.nanoen.2023.109180
- Han, H.; Huang, K.; Meng, X. Review on solar-driven evaporator: Development and applications. Ind. Eng. Chem2023, 119, 77-89. https://doi.org/10.1016/j.jiec.2022.11.051
- Chen, C.; Kuang, Y.; Hu, L. Challenges and Opportunities for Solar Evaporation. Joule2019, 3, 683-718. https://doi.org/10.1016/j.joule.2018.12.023
- Zhang, Q.; Xu, W.; Wang, X. Carbon nanocomposites with high photothermal conversion efficiency.Sci. China Mater. 2018, 61, 905-914. https://doi.org/10.1007/s40843-018-9250-x
- Zhu, H.; Jiang, X. Development of a General Fabrication Strategy for Carbonaceous Noble Metal Nanocomposites with Photothermal Property.Nanoscale Res. Lett 2020, 15. https://doi.org/10.1186/s11671-019-3242-1
- Huang, J.; Fu, J.; Li, L.; Ma, J. Mg-based metallic glass nanowires with excellent photothermal effect. Scr.Mater. 2023, 222, 115036. https://doi.org/10.1016/j.scriptamat.2022.115036
- Hasanzadeh, R.; Azdast, T.; Lee, P.C.; Park, C.B. A review of the state-of-the-art on thermal insulation performance of polymeric foams. Therm.Sci. Eng. Prog. 2023, 41, 101808. https://doi.org/10.1016/j.tsep.2023.101808
- Sakhadeo, N.N.; Patro, T.U. Exploring the Multifunctional Applications of Surface-Coated Polymeric Foams─A Review. Eng. Chem. Res. 2022, 61, 5366-5387.https://doi.org/10.1021/acs.iecr.1c04945
- Hu, X.; Yang, J.; Tu, Y.; Su, Z.; Guan, Q.; Ma, Z. Hydrogel-Based Interfacial Solar-Driven Evaporation: Essentials and Trails. Gels2024, 10. https://doi.org/10.3390/gels10060371
- Li, J.; Liu, Q.; He, J.; Zhang, Y.; Mu, L.; Zhu, X.; Yao, Y.; Sun, C.-L.; Qu, M. Aerogel-based solar interface evaporation: Current research progress and future challenges. Desalination2024, 569. https://doi.org/10.1016/j.desal.2023.117068
- Liu, X.; Tian, Y.; Caratenuto, A.; Chen, F.; Zheng, Y. Biomass‐Based Materials for Sustainably Sourced Solar‐Driven Interfacial Steam Generation. Energy Mater.2023, 25. https://doi.org/10.1002/adem.202300778
- Wang, S.-X.; Zhao, H.-B.; Rao, W.-H.; Huang, S.-C.; Wang, T.; Liao, W.; Wang, Y.-Z. Inherently flame-retardant rigid polyurethane foams with excellent thermal insulation and mechanical properties. Polymer 2018, 153, 616-625. https://doi.org/10.1016/j.polymer.2018.08.068
- Kirpluks, M.; Kalnbunde, D.; Benes, H.; Cabulis, U. Natural oil based highly functional polyols as feedstock for rigid polyurethane foam thermal insulation. Crop. Prod.2018, 122, 627-636. https://doi.org/10.1016/j.indcrop.2018.06.040
- An, W.; Sun, J.; Liew, K.M.; Zhu, G. Flammability and safety design of thermal insulation materials comprising PS foams and fire barrier materials.Materials & Design 2016, 99, 500-508. https://doi.org/10.1016/j.matdes.2016.03.080
- Shi, L.; Wang, Y.; Zhang, L.; Wang, P. Rational design of a bi-layered reduced graphene oxide film on polystyrene foam for solar-driven interfacial water evaporation. Mater. Chem. A 2017, 5, 16212-16219. https://doi.org/10.1039/c6ta09810j
- Nguyen, T.K.T.; Dao, Q.K.; Tanaka, D.; Nghiem, L.H.T.; Nguyen, M.V.; Nguyen, Z.H.; Pham, T.T. Flexible, affordable and environmentally sustainable solar vapor generation based on ferric tannate/bacterial cellulose composite for efficient desalination solutions. RSC Adv2021, 11, 31641-31649. https://doi.org/10.1039/d1ra05558e
- Kiriarachchi, H.D.; Hassan, A.A.; Awad, F.S.; El-Shall, M.S. Metal-free functionalized carbonized cotton for efficient solar steam generation and wastewater treatment. RSC Adv2021, 12, 1043-1050. https://doi.org/10.1039/d1ra08438k
- Wang, Y.; Zhao, L.; Zhang, F.; Yu, K.; Yang, C.; Jia, J.; Guo, W.; Zhao, J.; Qu, F. Synthesis of a Co-Sn Alloy-Deposited PTFE Film for Enhanced Solar-Driven Water Evaporation via a Super-Absorbent Polymer-Based "Water Pump" Design. ACS Appl. Mater.2021, 13, 26879-26890. https://doi.org/10.1021/acsami.1c02586
- Xiao, Y.; Wang, X.; Li, C.; Peng, H.; Zhang, T.; Ye, M. A salt-rejecting solar evaporator for continuous steam generation. Eeviron. Chem. Eng 2021, 9. https://doi.org/10.1016/j.jece.2020.105010
- Chang, C.; Liu, M.; Li, L.; Chen, G.; Pei, L.; Wang, Z.; Ji, Y. Salt-rejecting rGO-coated melamine foams for high-efficiency solar desalination. J .Mater. Res.2021, 37, 294-303. https://doi.org/10.1557/s43578-021-00328-w
- Lal, S.; Batabyal, S.K. Activated carbon-cement composite coated polyurethane foam as a cost-efficient solar steam generator. Clean. Prod 2022, 379. https://doi.org/10.1016/j.jclepro.2022.134302
- Chen, J.; Li, B.; Hu, G.; Aleisa, R.; Lei, S.; Yang, F.; Liu, D.; Lyu, F.; Wang, M.; Ge, X.; et al. Integrated Evaporator for Efficient Solar-Driven Interfacial Steam Generation. Nano Lett.2020, 20, 6051-6058. https://doi.org/10.1021/acs.nanolett.0c01999
- Lin, B.; Yuen, A.C.Y.; Oliver, S.; Liu, J.; Yu, B.; Yang, W.; Wu, S.; Yeoh, G.H.; Wang, C.H. Dual functionalisation of polyurethane foam for unprecedented flame retardancy and antibacterial properties using layer-by-layer assembly of MXene chitosan with antibacterial metal particles. Part BEng. 2022, 244, 110147. https://doi.org/10.1016/j.compositesb.2022.110147
- Pinto, J.; Magrì, D.; Valentini, P.; Palazon, F.; Heredia-Guerrero, J.A.; Lauciello, S.; Barroso-Solares, S.; Ceseracciu, L.; Pompa, P.P.; Athanassiou, A.; et al. Antibacterial Melamine Foams Decorated with in Situ Synthesized Silver Nanoparticles. ACS Appl. Mater. Interfaces2018, 10, 16095-16104. https://doi.org/10.1021/acsami.8b01442
- Wang, J.; Yang, W.; He, F.; Xie, C.; Fan, J.; Wu, J.; Zhang, K. Superhydrophobic Melamine-formaldehyde Foam Prepared by In-situ Coprecipitation. Lett2018, 47, 414-416. https://doi.org/10.1246/cl.171165
- Shi, H.-G.; Li, S.-L.; Cheng, J.-B.; Zhao, H.-B.; Wang, Y.-Z. Multifunctional Photothermal Conversion Nanocoatings Toward Highly Efficient and Safe High-Viscosity Oil Cleanup Absorption. ACS Appl. Mater. Interfaces 2021, 13, 11948-11957.https://doi.org/10.1021/acsami.0c22596
- Wang, Z.; Niu, J.; Wang, J.; Zhang, Y.; Wu, G.; Liu, X.; Liu, Q. Rational Design of Photothermal and Anti-Bacterial Foam With Macroporous Structure for Efficient Desalination of Water. Front Chem.2022, 10, 912489. https://doi.org/10.3389/fchem.2022.912489
- Gnanasekaran, A.; Rajaram, K. Rational design of different interfacial evaporators for solar steam generation: Recent development, fabrication, challenges and applications. Sust. Enegr. Rev. 2024, 192. https://doi.org/10.1016/j.rser.2023.114202
- Ullah, F.; Othman, M.B.H.; Javed, F.; Ahmad, Z.; Akil, H.M. Classification, processing and application of hydrogels: A review. Sci. Eng C 2015, 57, 414-433.https://doi.org/10.1016/j.msec.2015.07.053
- Jing, X.; Liu, F.; Abdiryim, T.; Liu, X. Hydrogels as promising platforms for solar-driven water evaporators. Eng. J.2024, 479. https://doi.org/10.1016/j.cej.2023.147519
- Zhao, Q.; Wen, H.; Wu, J.; Wen, X.; Xu, Z.; Duan, J. Galactomannan/graphene oxide/Fe3O4 hydrogel evaporator for solar water evaporation for synergistic photothermal power generation. Desalination2024, 570. https://doi.org/10.1016/j.desal.2023.117064
- Wang, J.; Guo, Z.; Xiao, B.; Xiong, X.; Liu, G.; Wang, X. Reduced graphene oxide/Cu72S4 composite hydrogels for highly efficient solar steam generation. Mater. Today. Sustain.2022, 18. https://doi.org/10.1016/j.mtsust.2022.100121
- Xiong, Y.; Hu, D.; Huang, L.; Fang, Z.; Jiang, H.; Mao, Q.; Wang, H.; Tang, P.; Li, J.; Wang, G.; et al. Ultra-high strength sodium alginate/PVA/PHMB double-network hydrogels for marine antifouling. Org. Coat.2024, 187, 108175. https://doi.org/10.1016/j.porgcoat.2023.108175
- Wang, B.; Zhu, H.; Shutes, B. Multi-strategy coupling of custom hydrogel evaporators for sustained, high-efficiency clean water production and anti-pollution. Nano Today2024, 57, 102370. https://doi.org/10.1016/j.nantod.2024.102370
- Chu, A.; Yang, M.; Chen, J.; Zhao, J.; Fang, J.; Yang, Z.; Li, H. Biomass-enhanced Janus sponge-like hydrogel with salt resistance and high strength for efficient solar desalination. Green Energy Environ. 2023. https://doi.org/10.1016/j.gee.2023.04.003
- Zhao, J.; Chu, A.; Chen, J.; Qiao, P.; Fang, J.; Yang, Z.; Duan, Z.; Li, H. Spongy polyelectolyte hydrogel for efficient Solar-Driven interfacial evaporation with high salt resistance and compression resistance. Eng. J.2024, 485, 150118. https://doi.org/10.1016/j.cej.2024.150118
- Peng, B.; Lyu, Q.; Gao, Y.; Li, M.; Xie, G.; Xie, Z.; Zhang, H.; Ren, J.; Zhu, J.; Zhang, L.; et al. Composite Polyelectrolyte Photothermal Hydrogel with Anti-biofouling and Antibacterial Properties for the Real-World Application of Solar Steam Generation. ACS Appl. Mater. Interfaces2022, 14, 16546-16557. https://doi.org/10.1021/acsami.2c02464
- Peng, B.; Gao, Y.; Lyu, Q.; Xie, Z.; Li, M.; Zhang, L.; Zhu, J. Cationic Photothermal Hydrogels with Bacteria-Inhibiting Capability for Freshwater Production via Solar-Driven Steam Generation. ACS Appl. Mater. Interfaces2021, 13, 37724-37733. https://doi.org/10.1021/acsami.1c10854
- Garg, S.; Singh, S.; Shehata, N.; Sharma, H.; Samuel, J.; Khan, N.A.; Ramamurthy, P.C.; Singh, J.; Mubashir, M.; Bokhari, A.; et al. Aerogels in wastewater treatment: A review. Taiwan. Inst. Chem. E2023, 105299, https://doi.org/10.1016/j.jtice.2023.105299
- S, S.S.; Rai, N.; Chauhan, I. Multifunctional Aerogels: A comprehensive review on types, synthesis and applications of aerogels. Sol-Gel. Sci. Techn.2023, 105, 324-336. https://doi.org/10.1007/s10971-022-06026-1
- Wu, J.; Yang, X.; Jia, X.; Yang, J.; Miao, X.; Shao, D.; Song, H.; Li, Y. Full biomass-derived multifunctional aerogel for solar-driven interfacial evaporation. Eng. J2023, 471. https://doi.org/10.1016/j.cej.2023.144684
- Storer, D.P.; Phelps, J.L.; Wu, X.; Owens, G.; Khan, N.I.; Xu, H. Graphene and Rice-Straw-Fiber-Based 3D Photothermal Aerogels for Highly Efficient Solar Evaporation. ACS Appl. Mater. Interfaces2020, 12, 15279-15287. https://doi.org/10.1021/acsami.0c01707
- Xiao, J.-K.; Gong, J.-Z.; Dai, M.; Zhang, Y.-F.; Wang, S.-G.; Lin, Z.-D.; Du, F.-P.; Fu, P. Reduced graphene oxide/Ag nanoparticle aerogel for efficient solar water evaporation. Alloy. Compd.2023, 930, 167404. https://doi.org/10.1016/j.jallcom.2022.167404
- Jian, H.; Wang, Y.; Li, W.; Ma, Y.; Wang, W.; Yu, D. Reduced graphene oxide aerogel with the dual-cross-linked framework for efficient solar steam evaporation. Surface A2021, 629, 127440. https://doi.org/10.1016/j.colsurfa.2021.127440
- Fillet, R.; Nicolas, V.; Fierro, V.; Celzard, A. A review of natural materials for solar evaporation. Energ. Mat. Sol. C 2021, 219. https://doi.org/10.1016/j.solmat.2020.110814
- Zhang, P.; Xie, M.; Jin, Y.; Jin, C.; Wang, Z. A Bamboo-Based Photothermal Conversion Device for Efficient Solar Steam Generation. ACS Appl. Mater. Interfaces2022, 4, 2393-2400. https://doi.org/10.1021/acsapm.1c01681
- Zhang, C.; Xiao, P.; Ni, F.; Yan, L.; Liu, Q.; Zhang, D.; Gu, J.; Wang, W.; Chen, T. Converting Pomelo Peel into Eco-friendly and Low-Consumption Photothermic Biomass Sponge toward Multifunctioal Solar-to-Heat Conversion.ACS Sustainable Chem. Eng. 2020, 8, 5328-5337. https://doi.org/10.1021/acssuschemeng.0c00681
- Chen, C.; Li, Y.; Song, J.; Yang, Z.; Kuang, Y.; Hitz, E.; Jia, C.; Gong, A.; Jiang, F.; Zhu, J.Y.; et al. Highly Flexible and Efficient Solar Steam Generation Device. Mater. 2017, 29. https://doi.org/10.1002/adma.201701756
- Song, D.; Zheng, D.; Li, Z.; Wang, C.; Li, J.; Zhang, M. Research Advances in Wood Composites in Applications of Industrial Wastewater Purification and Solar-Driven Seawater Desalination. Polymers 2023, 15.https://doi.org/10.3390/polym15244712
- Li, Y.; Li, Q.; Qiu, Y.; Feng, H. High-efficiency wood-based evaporators for solar-driven interfacial evaporation. Sol Energy2022, 244, 322-330. https://doi.org/10.1016/j.solener.2022.08.036
- Li, W.; Li, F.; Zhang, D.; Bian, F.; Sun, Z. Porous wood-carbonized solar steam evaporator. Wood Sci. Technol.2021, 55, 625-637. https://doi.org/10.1007/s00226-021-01270-0
- Chen, T.; Wu, Z.; Liu, Z.; Aladejana, J.T.; Wang, X.; Niu, M.; Wei, Q.; Xie, Y. Hierarchical Porous Aluminophosphate-Treated Wood for High-Efficiency Solar Steam Generation. ACS Appl. Mater. Interfaces2020, 12, 19511-19518. https://doi.org/10.1021/acsami.0c01815
- Fang, Q.; Li, T.; Chen, Z.; Lin, H.; Wang, P.; Liu, F. Full Biomass-Derived Solar Stills for Robust and Stable Evaporation To Collect Clean Water from Various Water-Bearing Media. ACS Appl. Mater. Interfaces2019, 11, 10672-10679. https://doi.org/10.1021/acsami.9b00291
- Huang, W.; Hu, G.; Tian, C.; Wang, X.; Tu, J.; Cao, Y.; Zhang, K. Nature-inspired salt resistant polypyrrole–wood for highly efficient solar steam generation. Energ. Fuels 2019, 3, 3000-3008. https://doi.org/10.1039/c9se00163h
- Shi, L.; Zhang, M.; Du, X.; Liu, B.; Li, S.; An, C. In situ polymerization of pyrrole on elastic wood for high efficiency seawater desalination and oily water purification. Mater. Sci. 2022, 57, 16317-16332. https://doi.org/doi:10.1007/s10853-022-07632-8
- Pham, T.T.; Nguyen, T.H.; Nguyen, T.A.H.; Pham, D.D.; Nguyen, D.C.; Do, D.B.; Nguyen, H.V.; Ha, M.H.; Nguyen, Z.H. Durable, scalable and affordable iron (III) based coconut husk photothermal material for highly efficient solar steam generation.Desalination 2021, 518, 115280. https://doi.org/10.1016/j.desal.2021.115280
- Long, S. Huang, H. Yi, J. Chen, J. Wu, Q. Liu, Carrot-inspired solar thermal evaporator.J. Mater. Chem. A 2019, 7 26911.
- Jia, X.; Liu, X.; Guan, H.; Fan, T.; Chen, Y.; Long, Y.-Z. A loofah-based photothermal biomass material with high salt-resistance for efficient solar water evaporation. Commun.2023, 37, 101430. https://doi.org/10.1016/j.coco.2022.101430
- Chen, T.; Xie, H.; Qiao, X.; Hao, S.; Wu, Z.; Sun, D.; Liu, Z.; Cao, F.; Wu, B.; Fang, X. Highly Anisotropic Corncob as an Efficient Solar Steam-Generation Device with Heat Localization and Rapid Water Transportation. ACS Appl. Mater. Interfaces2020, 12, 50397-50405. https://doi.org/10.1021/acsami.0c13845
- Xu, N.; Hu, X.; Xu, W.; Li, X.; Zhou, L.; Zhu, S.; Zhu, J. Mushrooms as Efficient Solar Steam-Generation Devices. Mater.2017, 29. https://doi.org/10.1002/adma.201606762
- Zeng, L.; Deng, D.; Zhu, L.; Wang, H.; Zhang, Z.; Yao, Y. Biomass photothermal structures with carbonized durian for efficient solar-driven water evaporation. Energy2023, 273, 127170. https://doi.org/10.1016/j.energy.2023.127170
- Zhang, Q.; Yang, X.; Deng, H.; Zhang, Y.; Hu, J.; Tian, R. Carbonized sugarcane as interfacial photothermal evaporator for vapor generation. Desalination2022, 526, 115544. https://doi.org/10.1016/j.desal.2021.115544
- Roy, A.; Tariq, M.Z.; La, M.; Choi, D.; Park, S.J. 3D carbonized orange peel: A self-floating solar absorber for efficient water evaporation. Desalination2024, 573, 117191. https://doi.org/10.1016/j.desal.2023.117191
- Li, Z.; Wei, S.; Ge, Y.; Zhang, Z.; Li, Z. Biomass-based materials for solar-powered seawater evaporation. Total. Environ.2023, 858, 160003. https://doi.org/10.1016/j.scitotenv.2022.160003
- Zhu, M.; Yu, J.; Ma, C.; Zhang, C.; Wu, D.; Zhu, H. Carbonized daikon for high efficient solar steam generation. Energ. Mat. Sol. C 2019, 191, 83-90.https://doi.org/10.1016/j.solmat.2018.11.015
- Guo, M.X.; Wu, J.B.; Zhao, H.Y.; Li, F.H.; Min, F.Q. Carbonized loofah andMOF‐801 of synergistic effect for efficient solar steam generation. J. Energ. Res. 2021, 45, 10599-10608. https://doi.org/10.1002/er.6547
- Zhang, W.; Zhang, L.; Li, T.; Wu, D.; Zhang, C.; Zhu, H. Efficient solar-driven interfacial water evaporation enabled wastewater remediation by carbonized sugarcane. Water. Process. Eng. 2022, 49. https://doi.org/10.1016/j.jwpe.2022.102991
- Cao, N.; Lu, S.; Yao, R.; Liu, C.; Xiong, Q.; Qin, W.; Wu, X. A self-regenerating air-laid paper wrapped ASA 3D cone-shaped Janus evaporator for efficient and stable solar desalination. Eng. J. 2020, 397. https://doi.org/10.1016/j.cej.2020.125522
- Zou, M.; Zhang, Y.; Cai, Z.; Li, C.; Sun, Z.; Yu, C.; Dong, Z.; Wu, L.; Song, Y. 3D Printing a Biomimetic Bridge-Arch Solar Evaporator for Eliminating Salt Accumulation with Desalination and Agricultural Applications. Mater. 2021, 33, e2102443.https://doi.org/10.1002/adma.202102443
- Chaule, S.; Hwang, J.; Ha, S.J.; Kang, J.; Yoon, J.C.; Jang, J.H. Rational Design of a High Performance and Robust Solar Evaporator via 3D‐Printing Technology. Mater.2021, 33. https://doi.org/10.1002/adma.202102649
- Khalil, A.; Ahmed, F.E.; Hilal, N. The emerging role of 3D printing in water desalination. Total. Environ. 2021, 790.https://doi.org/10.1016/j.scitotenv.2021.148238
- Wang, Y.; Wang, C.; Song, X.; Huang, M.; Megarajan, S.K.; Shaukat, S.F.; Jiang, H. Improved light-harvesting and thermal management for efficient solar-driven water evaporation using 3D photothermal cones. Mater. Chem. A2018, 6, 9874-9881. https://doi.org/10.1039/c8ta01469h
- Lv, F.; Miao, J.; Hu, J.; Orejon, D. 3D Solar Evaporation Enhancement by Superhydrophilic Copper Foam Inverted Cone and Graphene Oxide Functionalization Synergistic Cooperation.Small 2023, e2208137. https://doi.org/10.1002/smll.202208137
- Xie, M.; Zhang, P.; Cao, Y.; Yan, Y.; Wang, Z.; Jin, C. A three-dimensional antifungal wooden cone evaporator for highly efficient solar steam generation. NPJ Clean Water 2023, 6. https://doi.org/10.1038/s41545-023-00231-3
- Bu, Y.; Zhou, Y.; Lei, W.; Ren, L.; Xiao, J.; Yang, H.; Xu, W.; Li, J. A bioinspired 3D solar evaporator with balanced water supply and evaporation for highly efficient photothermal steam generation. Mater. Chem. A2022, 10, 2856-2866. https://doi.org/10.1039/d1ta09288j
- Xu, Z.; Ran, X.; Zhang, Z.; Zhong, M.; Wang, D.; Li, P.; Fan, Z. Designing a solar interfacial evaporator based on tree structures for great coordination of water transport and salt rejection.Mater. Horiz. 2023. https://doi.org/10.1039/d2mh01447e
- Xu, Z.; Ran, X.; Wang, D.; Zhong, M.; Zhang, Z. High efficient 3D solar interfacial evaporator: Achieved by the synergy of simple material and structure. Desalination2022, 525. https://doi.org/10.1016/j.desal.2021.115495
- Chen, Y.; Hou, R.; Yang, L.; Chen, C.; Cui, J.; Zhou, T.; Zhao, Y.; Song, J.; Fan, Z.; Tang, Y.; et al. Elastic, Janus 3D evaporator with arch-shaped design for low-footprint and high-performance solar-driven zero-liquid discharge. Desalination 2024, 583, 117644.https://doi.org/10.1016/j.desal.2024.117644
- Wu, L.; Dong, Z.; Cai, Z.; Ganapathy, T.; Fang, N.X.; Li, C.; Yu, C.; Zhang, Y.; Song, Y. Highly efficient three-dimensional solar evaporator for high salinity desalination by localized crystallization. Nat.Commun. 2020, 11, 521. https://doi.org/10.1038/s41467-020-14366-1
- Zhang, Z.; Feng, Z.; Qi, H.; Chen, Y.; Chen, Y.; Deng, Q.; Wang, S. Carbonized sorghum straw derived 3D cup-shaped evaporator with enhanced evaporation rate and energy efficiency. Mater. Techno. 2022, 32, e00414.https://doi.org/10.1016/j.susmat.2022.e00414
- Zhang, L.; Bai, B.; Hu, N.; Wang, H. Low-cost and facile fabrication of a candle soot/adsorbent cotton 3D-interfacial solar steam generation for effective water evaporation. Energ. Mat. Sol. C 2021, 221.https://doi.org/10.1016/j.solmat.2020.110876
- Sun, S.; Shi, C.; Kuang, Y.; Li, M.; Li, S.; Chan, H.; Zhang, S.; Chen, G.; Nilghaz, A.; Cao, R.; et al. 3D-printed solar evaporator with seashell ornamentation-inspired structure for zero liquid discharge desalination. Water Res.2022, 226. https://doi.org/10.1016/j.watres.2022.119279
- Zhang, X.; Yan, Y.; Li, N.; Yang, P.; Yang, Y.; Duan, G.; Wang, X.; Xu, Y.; Li, Y. A robust and 3D-printed solar evaporator based on naturally occurring molecules. Bull. 2023, 68, 203-213. https://doi.org/10.1016/j.scib.2023.01.017
- Chen, Y.; Yang, J.; Zhang, D.; Wang, S.; Jia, X.; Li, Y.; Shao, D.; Feng, L.; Song, H.; Tang, S. A wood-inspired bimodal solar-driven evaporator for highly efficient and durable purification of high-salinity wastewater. Mater. Chem. A2023, 11. 2349-2359. https://doi.org/10.1039/d2ta08275f
Thanks again for your carefully review and valuable comments, which improved the quantity of our manuscript highly! Sincerely thanks!
Reviewer 2 Report
Comments and Suggestions for Authors
sufficient changes have been incorporated.
Comments on the Quality of English Languagesufficient changes have been incorporated.
Author Response
Thanks again for your carefully review and valuable comments, which improved the quantity of our manuscript highly! Sincerely thanks!